# Resolving dynamics and function of transient states in single enzyme molecules

Hugo Sanabria [1,2,8✉], Dmitro Rodnin[1,8], Katherina Hemmen [1,8], Thomas-Otavio Peulen [1], Suren Felekyan [1], Mark R. Fleissner[3,7], Mykola Dimura [1,4], Felix Koberling[5], Ralf Kühnemuth[1], Wayne Hubbell[3], Holger Gohlke [4,6] & Claus A.M. Seidel [1✉]

We use a hybrid fluorescence spectroscopic toolkit to monitor T4 Lysozyme (T4L) in action by unraveling the kinetic and dynamic interplay of the conformational states. In particular, by combining single-molecule and ensemble multiparameter fluorescence detection, EPR spectroscopy, mutagenesis, and FRET-positioning and screening, and other biochemical and biophysical tools, we characterize three short-lived conformational states over the ns-ms timescale. The use of 33 FRET-derived distance sets, to screen available T4L structures, reveal that T4L in solution mainly adopts the known open and closed states in exchange at 4 µs. A newly found minor state, undisclosed by, at present, more than 500 crystal structures of T4L and sampled at 230 µs, may be actively involved in the product release step in catalysis. The presented fluorescence spectroscopic toolkit will likely accelerate the development of dynamic structural biology by identifying transient conformational states that are highly abundant in biology and critical in enzymatic reactions.

[1] Institut für Physikalische Chemie, Lehrstuhl für Molekulare Physikalische Chemie, Heinrich-Heine-Universität, Düsseldorf, Germany. [2] Department of Physics and Astronomy, Clemson University, Clemson, SC, USA. [3] Jules Stein Eye Institute and Department of Chemistry and Biochemistry, University of California, Los Angeles, CA, USA. [4] Institut für Pharmazeutische und Medizinische Chemie, Heinrich-Heine-Universität, Düsseldorf, Germany. [5] PicoQuant GmbH, Berlin, Germany. [6] John von Neumann Institute for Computing (NIC), Jülich Supercomputing Centre (JSC) & Institute of Biological Information Processing (IBI-7: Structural Biochemistry), Forschungszentrum Jülich GmbH, 52425 Jülich, Germany. [7] Present address: Avanir Pharmaceuticals Inc., Aliso Viejo, CA, USA. [8] These authors contributed equally: Hugo Sanabria, Dmitro Rodnin, Katherina Hemmen. ✉email: hsanabr@clemson.edu; cseidel@hhu.de

Enzymes adopt distinct conformational states during catalysis[1,2], where transiently populated ("excited") states are often of critical importance in the enzymatic cycle. These states are short-lived and therefore "hidden" to many experimental techniques. Classical structural biology methods often struggle to fully capture enzymes during catalytic action because the conformational rearrangements often span several decades in time (ns-ms)[3–8]. Hence, there is an urgent need to develop experimental and analysis methods to overcome this challenge. Recently, we demonstrated by simulated experiments that a new analysis toolkit ("FRET on rails") combined with molecular simulations can resolve short-lived conformational states of proteins[9].

Here, we apply and extend the fluorescence analysis toolkit[9,10], developed for dynamic structural biology, to interrogate the catalytic cycle of an enzyme[11]. In particular, the analysis (1) captures an excited, short-lived state and (2) identifies its potential relevance in the enzyme's catalytic cycle. The presented approach may serve as a blueprint for future enzymologic studies with the well-established single-molecule multiparameter fluorescence detection (MFD) experiments in that it enables detecting hidden states by the unique time-resolution (picoseconds) and sensitivity (single-molecule) of fluorescence.

We use lysozyme (T4L) of the bacteriophage T4 as development platform and probe its conformational dynamics and structural features. Structurally, T4L[12] consists of two interrelated subdomains, the N-terminal subdomain (NTsD) and the C-terminal subdomain (CTsD), differing in their folding behavior and stability[13]. A long α-helix (helix c) links the two subdomains (Fig. 1a). To date, more than 500 structural models of T4L are available within the Protein Data Bank (PDB). In this ensemble, T4L adopts several opening angles corresponding to a classic hinge-bending motion of the NTsD with respect to the CTsD. The enzymatic function of T4L is to cleave the glycosidic bond between N-acetylmuramic acid and N-acetylglucosamine of the saccharides of the bacterial cell wall[14].

T4L in solution is thought to adopt conformations that are open to various degrees, and a covalent adduct of the protein and its processed enzymatic product can crystallize in a closed conformation[14–16]. Therefore, T4L is thought to follow a classical Michaelis–Menten mechanism (MMm) characterized as a two-state system (Fig. 1b). Here, an open and closed conformational state fulfils unique functions of substrate binding and substrate cleavage, respectively[17]. In a MMm, the product dissociates stochastically from the enzyme. For other enzymes, e.g. the Horseradish peroxidase[18], an "active" product release state was identified. Recent experimental findings for T4L suggest the involvement of more than two states in catalysis[19,20], where the turnover rate was estimated between 10 and 50 ms[20–23], while the conformational dynamics fell within the ns to sub-ms range[4,15,20–29]. Such complex cases, with distinct interconverting conformational states, open additional reaction paths and yield disperse kinetics[30].

For a full description of an enzymological cycle, the number of enzymatic states, their connectivity, the conformational structures of the states, and the states' chemical function have to be unraveled. Technically, we achieve these objectives by a hybrid approach combining classic biochemical methods (mutagenesis & HPLC), probe-based spectroscopy, and molecular simulations. Förster resonance energy transfer (FRET) and electron paramagnetic resonance (EPR) spectroscopy probe distances between bioorthogonally introduced probes through dipolar coupling. In FRET spectroscopy, the coupling is measured between a donor (D) and acceptor (A) fluorophore.

In confocal MFD single-molecule FRET (smFRET) experiments, freely diffusing molecules are repeatedly excited by a pulsed light source, and the emitted fluorescence photon is detected with picosecond time-resolution by time-correlated single photon counting (TCSPC) for several milliseconds per molecule (diffusion time, $t_{\text{diff}}$)[31] (Fig. 1c). smFRET experiments are ideal to study kinetics because no sophisticated strategies are necessary to synchronize molecules prior to the analysis. Consequently, it is possible to probe reliably protein kinetics over seven decades in time (sub ns-ms).

Distinct features of photon streams are highlighted by different representations (Fig. 1d). (1) A MFD-histogram is particularly valuable to reveal the number of states, identify dynamics, and to inform on state connectivities. A MFD-histogram is generated by analyzing two complementary FRET-indicators, the average intensity-based FRET-efficiency, $E$, and the fluorescence-averaged donor lifetime in the presence of acceptor, $\langle \tau_{\text{D(A)}} \rangle_{\text{F}}$, for individual single-molecule events[31–33]. (2) Filtered fluorescence correlation spectroscopy (fFCS) quantifies exchange dynamics among the states by determining relaxation times[34,35]. (3) The analysis of fluorescence decays reveals populations of states and equilibrium distance distributions. (4) Finally, these experimental distances can be translated to structural models by molecular simulations[10,36,37].

Following the concepts that were established for simulated data to resolve the structure and dynamics of proteins by integrative studies with FRET experiments[8,9] (Fig. 1e and more detailed in Supplementary Fig. 1), we start out with a systematic design of a FRET network for T4L to simultaneously monitor its dynamics and structural features. In step II, we use a combination of MFD to resolve the conformational states stable at the ns timescales. In step III, we quantify the conformational dynamics by employing fFCS and Monte Carlo simulations to resolve the connectivity of the conformational states. Following this, we perform a statistical analysis to further substantiate the existence of a hidden conformational state of T4L that was clearly identified above. In step IV, we identify structural models by using three distinct distance sets to screen an ensemble of structural models and to compare our identified states with models in the PDB. In the final step V, we derive an experimental energy landscape of T4L's enzymatic cleavage cycle, based on shifting equilibrium, by mutating key active site residues that mimic functional enzyme states at various steps during substrate hydrolysis. Overall, our results demonstrate the potential of fluorescence spectroscopy to go beyond traditional experimental methods for obtaining a dynamic structural picture of enzymes in action. The existence of the identified hidden/excited conformational state is also corroborated by other analytical tools such as chromatography and EPR spectroscopy.

## Results

**Detecting T4L's states by MFD.** In our smFRET-experiments, we monitor the distance between a donor (D) and acceptor (A) attached to specific amino acids of a T4L variant (see Methods). We designed a network of 33 distinct T4L variants to probe hinge-bending motions of T4L from different spatial directions (Fig. 2a) that cover the whole protein.

In Fig. 2b, c, we present MFD-histograms with the two FRET-indicators for two exemplary variants of our FRET network. Three peaks are identified in the MFD-histogram. In both histograms, a major and a minor FRET peak are present. The peak located at a low FRET-efficiency $E$ corresponds to molecules without, or with an inactive acceptor, fluorophore (DOnly).

For an open (PDB ID: 172L, blue) and a closed (PDB ID: 148L, magenta) conformation, FRET efficiencies $E$ are predicted by experimentally calibrated dye models (see Methods, section 5.5)[9,38]. These $E$ values are shown as horizontal lines in the marginal distributions of Fig. 2b, c. A comparison with the major peak (Fig. 2b: $0.3 < E < 0.7$, Fig. 2c: $E > 0.7$) demonstrates that they are

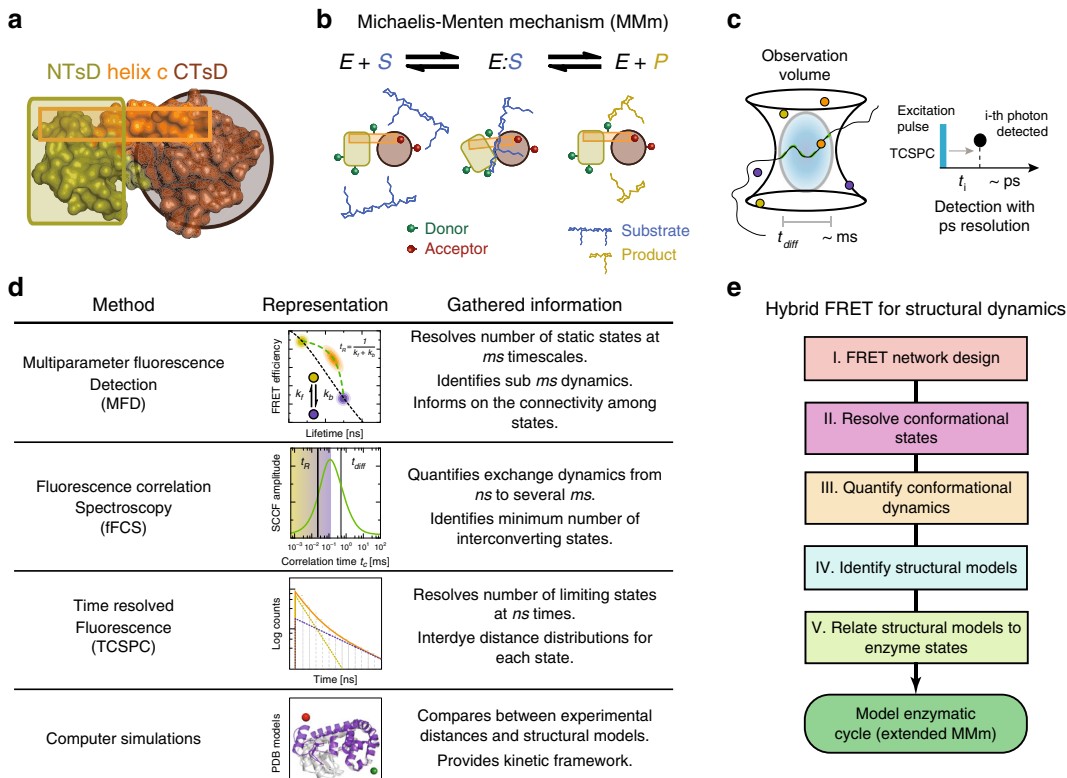

**Fig. 1 Schematic of the high precision FRET and fluctuation analysis toolkit. a** Subdomain architecture of Phage T4 Lysozyme (T4L). **b** Upon cleavage of its substrate peptidoglycan (blue), the N-terminal lobe (green) and the C-terminal lobe (brown) of T4L exhibit a change of closure, which can be observed via the induced change in the FRET indicators[14,15]. **c** In MFD-experiments of freely diffusing single molecules, the emitted fluorescent photons ("bursts") are detected with ps resolution (with respect to the exciting laser pulse) during the diffusion (on the ms time scale) of the molecule through the observation volume (diffusion time, $t_{diff}$). **d** The labeled T4L molecules are studied with different experimental methods. In single-molecule multiparameter fluorescence detection (MFD) experiments, the fluorescence bursts—averaged over ms—are analyzed e.g. with respect to their fluorescence lifetime $\langle\tau_{D(A)}\rangle_F$ or FRET-efficiency $E$ and displayed in multidimensional frequency histograms (2D MFD-histogram). Molecules that adopt a stable conformation during burst duration are located along the static FRET-line (black) (Supplementary Methods). Assuming that the two limiting states (yellow and blue) exchange on timescales faster than ms with exchange rate constants $k_f$ and $k_b$, we find only a single population (orange) shifted towards a longer fluorescence lifetime that is located on the dynamic FRET-line (green) connecting these two limiting states. Thus, FRET-lines serve as a visual guide to interpret 2D MFD-histograms, with deviations from the static FRET-line being indicative for the dynamic averaging and dynamics at the sub-ms and ms timescales. Filtered fluorescence correlation spectroscopy (fFCS) computes the species-specific cross-correlation function (sCCF (green), Supplementary Methods). The observed anti-correlation reveals a characteristic relaxation time $t_R$ related to the inverse of the sum of the exchange rate constants, $k_f$ and $k_b$. In eTCSPC, the distribution of the fluorescent photons (yellow/blue—individual states, orange—mixture) is detected with respect to an excitation pulse with ps resolution to reveal populations stable on the ns timescale (Supplementary Methods). Finally, in molecular simulations, the experimental results are compared to available structural models. **e** Flowchart for the hybrid FRET toolkit for determining structural dynamics. Based on a network of FRET variants, the conformational states and their exchange dynamics are determined, which are then used to identify the structural models. T4L variants with mutations altering their enzymatic activity relate the structural models to enzymatic states. Based on the gathered information, the enzymatic cycle can be modeled.

similar but not identical to known structural models. Next, we will show how dynamic exchange explains the observed peaks.

In MFD-histograms, FRET-lines (Supplementary Methods) serve as a unique guide to visualize conformational dynamics by peak shifts and splitting like in NMR relaxation dispersion measurements. A static FRET-line relates $E$ and $\langle\tau_{D(A)}\rangle_F$ for molecules in the absence of dynamics (magenta line, Fig. 2b, c). States that exchange on a time scale much slower than the observation time (quasi-static case) are separated in a MFD-histogram and follow the static FRET-line. However, a shift of a peak to the right with respect to the static FRET-line is a model-free indication for sub-ms dynamics[31], because the FRET-efficiencies in a MFD-histogram are averaged over the observation time of the molecules (~ms). Thus, very fast exchanging states result in a single average peak that is shifted to the right of the static FRET-line. These peaks can be described by dynamic FRET-lines, which connect the exchanging states. For a visual representation of the possible transitions, the dynamic FRET-

lines of the identified exchanging states are displayed in the MFD-diagrams (dark green, cyan, and light green). A dynamic FRET-line connecting high FRET states with the DOnly population (gray) demonstrates the lack of significant photobleaching or blinking of acceptor dyes.

In the presented data, the major populations are shifted to the right of the static FRET-line (Fig. 2b, c). This gives clear evidence for a dynamic exchange faster than ms. For molecules in very rapid (µs) exchange between an open and a closed conformation, we expected to detect a single averaged peak in MFD-histograms. Hence, taking fast exchange into account, the major peak of the smFRET data is in agreement with known X-ray structures[14,39] and kinetic data[4,13,20,21,23,24,29,40–42], most likely corresponding to the dynamic averaging of the hinge bending mechanism.

However, in 18 out of 33 MFD-histograms, we visually identify additional minor populations, which are in slow exchange with the major populations. Surprisingly, these minor populations ($E > 0.8$, Fig. 2b) and ($0.2 < E < 0.6$, Fig. 2c) can neither be assigned to the

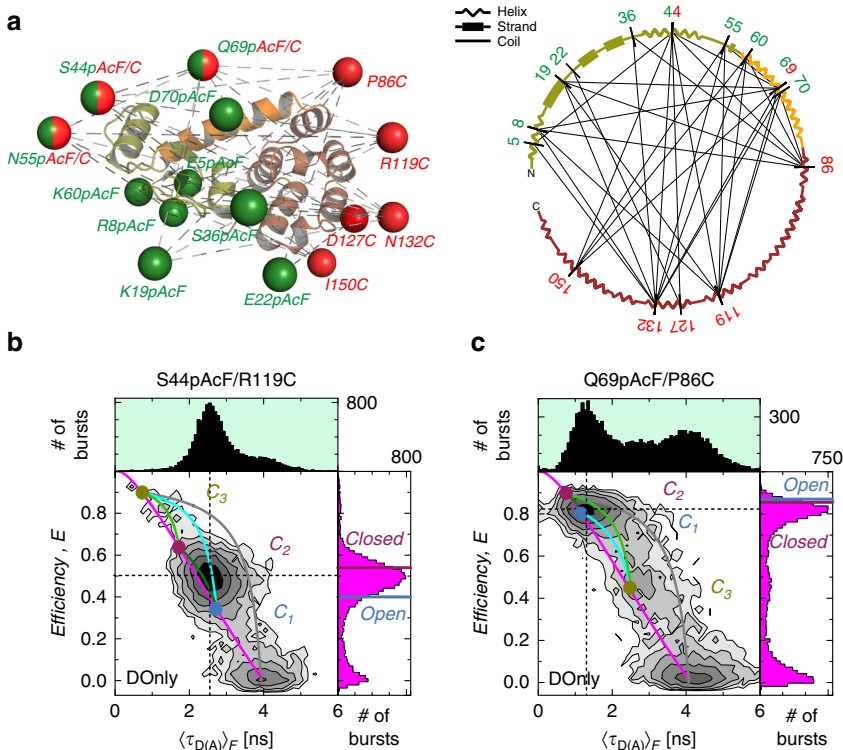

**Fig. 2 FRET studies probe T4L structure and dynamics. a** Network representing 33 measured distinct FRET-variants of T4L. The donor (D) Alexa488-hydroxylamine and acceptor (A) Alexa647-maleimide are coupled to *para*-acetylphenylalanine (pAcF) and cysteine (C), respectively. The spheres in the left panel represent the average donor (green) and acceptor (red) position for a structure of T4L (PDB ID: 172L) that is determined by the FRET Positioning System (FPS)[10]. Positions 44, 55, and 69 are used for donor and acceptor labeling. Right panel shows the secondary structure elements (helix, strand or coil) of T4L. The labeling positions are indicated and network pairs are connected. The N-terminal lobe is shown in green, the connecting helix c in orange and the C-terminal lobe in brown. **b, c** The FRET-efficiency, $E$, and fluorescence lifetime of D in the presence of A, $\langle\tau_{D(A)}\rangle_F$, of the DA-labeled T4L variants (**b**) S44pAcF/R119C and (**c**) Q69pAcF/P86C are shown in two-dimensional MFD-histograms (center) with one-dimensional projections of $E$ (right) and $\langle\tau_{D(A)}\rangle_F$ (top). The magenta lines (static FRET-line) describe those molecules with a single conformational state. The limiting states (circles) of the dynamic FRET-lines ($C_1$, blue; $C_2$, purple; $C_3$, yellow) are identified by eTCSPC (Supplementary Methods, Supplementary Note 4). The dynamic FRET-lines are shown in dark green ($C_1$–$C_2$), cyan ($C_1$–$C_3$), and light green ($C_2$–$C_3$) (Supplementary Methods). The gray lines trace molecules of high FRET-efficiency with a bleaching A. For comparison, the FRET-efficiencies of the X-ray structures for an *open* (blue, PDB ID: 172L) and closed (violet, 148L) state (determined by FPS) are shown as horizontal lines in the FRET-efficiency histograms (magenta) in the left 1D projection; for S44pAcF/R119C $E$(open) = 0.40 and $E$(closed) = 0.54 and for Q69pAcF/P86C $E$(open) = 0.87 and $E$(closed) = 0.86. Black dashed lines mark the center of the main peak.

predicted average open and closed conformation (Supplementary Fig. 2). This is a first indicative for the existence of a third, conformationally excited, structurally distinct conformer.

In conclusion, MFD-histograms identify three conformers in T4L referred to as $C_1$, $C_2$, and $C_3$. The conformers $C_1$ and $C_2$ are likely in fast exchange, while $C_3$ is in slow exchange with $C_1$ or $C_2$. These conformers may represent limiting states in the exchange dynamics[43,44].

Following the workflow presented in Fig. 1e, we next determine the kinetic signatures via fFCS and resolve remaining ambiguities by simulations of MFD-experiments.

**Connectivity of states in a kinetic network**. To construct a reaction scheme of T4L's enzymatic cycle, the variant S44pAcF/I150C is used as pseudo-wildtype ("wt\*\*"). At first, we carry out control experiments by comparing for this variant (DA)-labeled and reversely (AD)-labeled T4L variants. In this way, we could exclude potential dye artifacts (Supplementary Fig. 3a–c, Supplementary Note 1) because the kinetic behavior was independent of the labeling scheme. The MFD-histogram of S44pAcF/I150C (Fig. 3a) reveals a typical pattern: a major population $C_1/C_2$ ($0.2 < E < 0.6$) and a minor $C_3$ population ($E > 0.8$) similar to the variants presented in Fig. 2b, c.

To unravel the kinetic behavior of an enzyme, one has to be aware that an enzymatic cycle with multiple states can be described by a transition rate matrix, which contains all exchange rate constants of the states. To recover T4L's transition rate matrix, we determine a set of relaxation times by fFCS (see next paragraph) and the species fractions of the states by analysis of the fluorescence decays (for details see the section below). This analysis results in ambiguous solutions, which are resolved by simulating MFD experiments making use of the information contained in smFRET experiments.

**Kinetic network of conformational states resolved by fFCS**. By fFCS, we probe transitions in T4L on all relevant timescales[34,35] to resolve the kinetic network of conformational states. fFCS uses species-specific information encoded as a characteristic pattern within the ns-regime of the polarization-resolved fluorescence decays[34,45]. This amplifies the contrast compared to conventional FCS for resolving relaxation times with high precision. We find very good agreement between the normalized species cross-correlation functions (sCCF) of the (AD)- and (DA)-labeled molecules. A global analysis of the sCCFs and the species auto-correlation functions (sACF) requires at least two relaxation times ($t_{R1} = 4$ μs and $t_{R2} = 230$ μs, Fig. 3b, Supplementary Note 2).

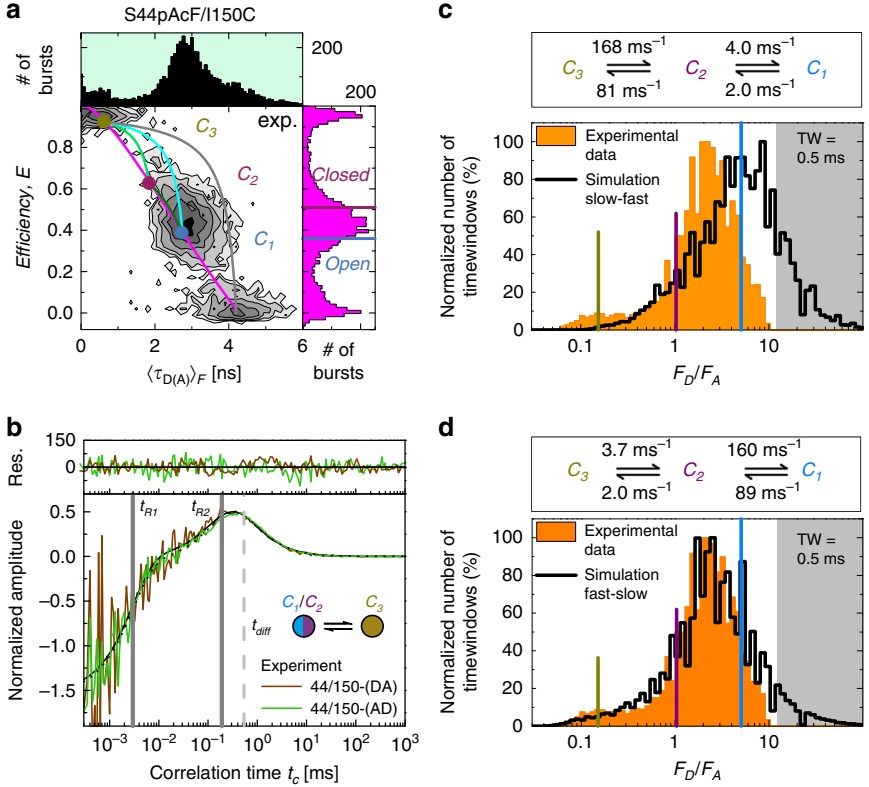

**Fig. 3 Kinetic connectivity. a** The FRET-efficiency, $E$, and fluorescence lifetime of D in the presence of A, $\langle\tau_{D(A)}\rangle_F$, of the DA-labeled T4L variant S44pAcF/
I150C (or wt**) is shown in the two-dimensional MFD-histograms (center) with the one-dimensional projections of $E$ (right) and $\langle\tau_{D(A)}\rangle_F$ (top). The magenta
line (static FRET-line) represents the molecules with a single conformation. The dynamic FRET-lines represent the molecules changing their conformational
state under monitoring. The limiting states (circles) of the dynamic FRET-lines ($C_1$, blue; $C_2$, purple; $C_3$, yellow) are identified by eTCSPC (see Fig. 4). The
dynamic FRET-lines are shown in dark green ($C_1$–$C_2$), cyan ($C_1$–$C_3$), and light green ($C_2$–$C_3$) (Supplementary Methods). The gray line represents the trace
molecules of high FRET-efficiency with a bleaching A. The FRET-efficiencies of X-ray structures for an open (blue, PDB ID: 172L) and closed (violet, 148L)
state (determined by FPS) are shown for comparison and represented as horizontal lines in the FRET-efficiency histograms. **b** Overlay of the normalized
sCCFs (Supplementary Methods Equation (17)) of S44pAcF/I150C-(DA) and S44pAcF/I150C-(AD). The global fit with other variants shows two relaxation
times ($t_{R1} = 4.0 \pm 2.3$ µs, $t_{R2} = 230 \pm 28$ µs) and a diffusion time $t_{diff} = 0.54$ ms. **c, d** Overlay of time-window analysis, slow-fast in **c** and fast-slow in **d**,
respectively. The shaded area in gray indicates the region of donor only. See Supplementary Note 3 for data on the model and the simulation.

In summary, the two relaxation times obtained by *sCCFs*
independently support the hypothesis of the interconversion
between three states at sub-ms timescales. Moreover, in line with
the MFD-histograms, we find a fast and a slow relaxation time.

Simulation of the kinetic network of T4 Lysozyme. The three
identified conformers $C_1$, $C_2$, and $C_3$ are assigned by their
characteristic species fractions (see below) to the corresponding
structural states open, closed, and excited, respectively.

Three distinct kinetic linear reaction schemes are possible:
$C_1 \leftrightarrows C_2 \leftrightarrows C_3$, $C_2 \leftrightarrows C_1 \leftrightarrows C_3$, $C_1 \leftrightarrows C_3 \leftrightarrows C_2$ while the cycle
scheme $C_1 \leftrightarrows C_2 \leftrightarrows C_3 \leftrightarrows C_1$ is unlikely due to the lack of burst
across all FRET variants that connect $C_3 \leftrightarrows C_1$ with an effective
slower rate to satisfy equilibrium. These bursts would follow the
dynamic FRET-lines as guides between states in the MFD-
diagram (Fig. 3a and Supplementary Fig. 2). Nonetheless, the
sequential closing, from the most open (lowest FRET-efficiency in
the variant S44pAcF/150C) state to the most compact (highest
FRET-efficiency in the variant S44pAcF/150C) is depicted by
$C_1 \leftrightarrows C_2 \leftrightarrows C_3$, so that we can discard the models $C_1 \leftrightarrows C_3 \leftrightarrows C_2$
and $C_2 \leftrightarrows C_1 \leftrightarrows C_3$. With the relaxation times determined by fFCS
and the species fractions obtained by analysis of the fluorescence
decays (see next section), we calculate the exchange rate constants
and find two competing solutions (Supplementary Note 3,
Equation (36)). The exchange between $C_1$ and $C_2$ can be either
slow (Fig. 3c) or fast (Fig. 3d).

To solve this ambiguity, we simulate sm-experiments of the
two possible solutions[43] (Supplementary Note 3). The obtained
simulations are compared with experimental histograms and
fFCS (Fig. 3c, d, Supplementary Fig. 4). The corresponding *p*-
values ($C_1$–$C_2$ fast vs. $C_1$–$C_2$ slow) of the respective 1D ($p = 1$ for
$\langle\tau_{D(A)}\rangle_F$, $p = 0.734$ for $E$) and 2D MFD-histograms ($p = 1$) clearly
demonstrate a better agreement of the experimental data with a
fast exchange between $C_1$ and $C_2$ (Supplementary Note 3,
Supplementary Table 1a, b).

To conclude, we experimentally determine all reaction rate
constants that define the reaction network, and the resulting
species fractions. This description covers µs – ms and captures the
relevant global motions of T4L.

**Characterization of the third conformer by eTCSPC.** As
demonstrated by fFCS analysis, T4L is highly dynamic. Hence,
the FRET-efficiencies in smFRET-histograms only represent
dynamic averages of states[46]. Therefore, for resolving the lim-
iting states of the system, we record high-precision fluorescence
decays by eTCSPC and analyze the distribution of the photon
arrival times, $t$, with respect to the excitation pulse in fluores-
cence decays. This analysis benefits from polarization-free
effects resulting from measuring at magic-angle detection, low
background fluorescence, and the absence of photobleaching.

Moreover, it can reveal DA-distance distributions and species populations[47]. To dissect the donor quenching by FRET (i.e., FRET-induced donor decay), we jointly analyze the DA- and DOnly-dataset, where the fluorescence lifetime distribution is shared with the DA-dataset. For physically meaningful analysis results, we explicitly consider the DA-distribution broadening due to the linkers by normal distributions[47].

The analysis results of all 33 FRET-datasets are discussed using the variant S36pAcF/P86C shown in Fig. 4a (for other variants see Supplementary Note 4, Supplementary Fig. 5a). We display the experimental data by fluorescence decays of the DA- and the corresponding DOnly-sample (Fig. 4a). In agreement with the MFD-histograms and the fFCS data, 1-component models result in broad DA-distributions and/or are insufficient to describe the data (Fig. 4a, weighted residuals, violet). For S36pAcF/P86C, we obtain both, an unphysical distribution width and significant deviations in the weighted residuals, a strong indication that more than one conformer is found.

The analysis of the fluorescence decays by a 2-component model yields an inconsistent assignment by the species fractions (Supplementary Fig. 5b, c). This is evident by significant differences among the species fractions (Supplementary Table 2b). Moreover, the DA-distances disagree with known structural models (compare Supplementary Table 2d, e).

In our effort to seek a consistent description of all measured fluorescence decays, we develop a joint/global model function. For such description, we treated all fluorescence decays as a single dataset sharing common species fractions for the states. This reduces the number of free parameters and dramatically stabilizes the optimization algorithm. Because the global 2-component model (Fig. 4, cyan, Supplementary Table 2c) shows no agreement with the data, we consequently used a 3-component model (Fig. 4a orange, Supplementary Table 2d–f) to describe the data.

To analyze the precision of this fit, the uncertainties $\Delta R_{DA}$ of the obtained distances, $\langle R_{DA} \rangle$, from the 3-component model need to be determined. $\Delta R_{DA}$ depends on statistical uncertainties and systematic errors. We use the known shot noise of the fluorescence decays to estimate the statistical uncertainties, $\Delta R_{DA}(k_{FRET})$, of the FRET-rate constant $k_{FRET}$ (Fig. 4b, Supplementary Table 2g). Moreover, we record polarization-resolved fluorescence decays of the donor and acceptor by eTCSPC to analyze the time-resolved anisotropy (Supplementary Table 3a, b) for estimating systematic errors, $\Delta R_{DA}(\kappa^2)$, due to the orientation factor $\kappa^2$. In conclusion, we can demonstrate that $\Delta R_{DA}(\kappa^2)$ dominates the overall uncertainty of $\Delta R_{DA}$ (Eq. (5), Supplementary Table 2d–g).

Moreover, we sample the model parameters of a 3-component model for individual datasets by a Markov chain Monte Carlo (MCMC) method. This demonstrates that, for given state populations, the mean distances $\langle R_{DA,1} \rangle$, $\langle R_{DA,2} \rangle$, and $\langle R_{DA,3} \rangle$ are very well defined (compare red to black in Fig. 4b). This also shows that a global model, which interrelates the state populations among datasets, improves the capability to resolve interdye distances.

A global 3-component model has too many degrees of freedom (Supplementary Methods) to be exhaustive when sampling by MCMC. Hence, we vary the state population of the minor state, $x$ ($C_3$), while optimizing all other model parameters (support plane analysis). This way, we determine the dependency of $x(C_3)$ on the quality parameter $\chi^2_r$ of all measurements (Fig. 4c). This analysis (1) shows that the minor state population is in the range of 0.1–0.27 and best agrees with the data for 0.21 (Fig. 4c, p-value = 0.68), (2) provides an estimate for $\Delta R_{DA}(k_{FRET})$ (Supplementary Table 2d–f), and (3) demonstrates that $\Delta R_{DA}(\kappa^2)$ dominates $\Delta R_{DA}$ (Eq. (5)).

In summary, only a 3-component analysis describes all FRET samples and reference samples in a global model. This analysis recovers a set of physically meaningful average DA-distances that are grouped automatically and unbiased by their state populations. Additionally, the 3-component model is consistent with the fFCS data and with the dynamic FRET-lines displaying dynamically averaged sm-subpopulations in MFD (Fig. 2, Supplementary Fig. 2).

The integrated results are consistent with a view that T4L adopts three states ($C_1$, $C_2$, and $C_3$), as opposed to the expected two conformational states based on structural pre-knowledge.

**Structural features of conformational states**. To compare the experimental distances $\langle R_{DA,exp} \rangle$ obtained from the fluorescence decays—under consideration of their respective uncertainties $\Delta R_{DA}$—to the structural models deposited in the PDB, we cluster all available 578 structures of T4L and aligned them. We observed that the structural models of T4L group into open, ajar, and closed clusters (based on the proximity of the CTsD and NTsD, Supplementary Table 4) with an intra-cluster root mean-squared displacement of less than 1.8 Å. The representative structures of these clusters are given by PDB IDs 172L, 1JQU, and 148L for the open, ajar, and closed conformations, respectively (Fig. 5a).

Next, we apply the FRET positioning system (FPS)[34] to compute an error function ($\chi^2_{r,FPS}$) that compares the three sets of 33 distances $\langle R_{DA,exp} \rangle$ to the modeled distances $\langle R_{DA,model} \rangle$ by FPS. In $\chi^2_{r,FPS}$, we consider explicitly the uncertainties, $\Delta R_{DA}$, of the distance $\langle R_{DA,exp} \rangle$[9]. The overall agreement (minimum $\chi^2_{r,FPS}$) for the distance sets for $C_1$ and $C_2$ is best for 172L and 148L, respectively (Fig. 5b). In Fig. 5c, $\langle R_{DA,model} \rangle$ for 172L and 148L are compared to $\langle R_{DA,exp} \rangle$ of $C_1$ and $C_2$, respectively. A linear regression (red line) with a slope close to one demonstrates the absence of significant systematic deviations.

Structurally, the ajar state is more closed than the open state and more open than the closed state, most likely representing an intermediate conformation or it could arise from structural instabilities introduced by specific mutations such as W158L[48]. The deviation from the open and closed state is clearly reflected in the elevated $\chi^2_{r,FPS}$. Consequently, within our precision we can safely assign $C_1$ as open and $C_2$ as closed state. Screening results of other structures in the PDB against the FRET data are very similar to the results for the discussed cluster representatives, as expected (Supplementary Fig. 6). However, none of the structures can be assigned to the $C_3$ state as judged by the disagreement with the data (Fig. 5b, $\chi^2_{r,FPS}$). Thus, we conclude that $C_3$ is an excited conformational state of currently unknown structure.

**Relevant functional states in the enzymatic cleavage cycle.** Detection of $C_3$ by EPR. We use double electron-electron resonance (DEER) to provide additional support for the $C_3$ state. Multiple DEER studies on T4L have shown interspin distributions for wt T4L[49,50]. Here, we show the distribution of interspin distances of the adduct form of the variant T26E/S44pAcF/I150C labeled with the appropriate spin label MTSSL to produce the variant T26E(+)- 44R1/150R1, which displays a satellite population with interspin distance of ~35 Å resembling the enzyme-product-complex $EP$ within the catalytic cleavage cycle of T4L (Fig. 6a, Supplementary Fig. 5d). The most frequently observed distance falls at interspin distances of 42 Å with another less populated state at interspin distances of >50 Å. These may correspond to the various sub-states of the closed ($C_2$) and open ($C_1$) states, respectively. To ensure that this small population is not an artifact of the Tikhonov regularization algorithm[51,52] or due to the rotamer populations of the spin label-carrying side chain, we lower the pH to influence the conformational equilibrium of the

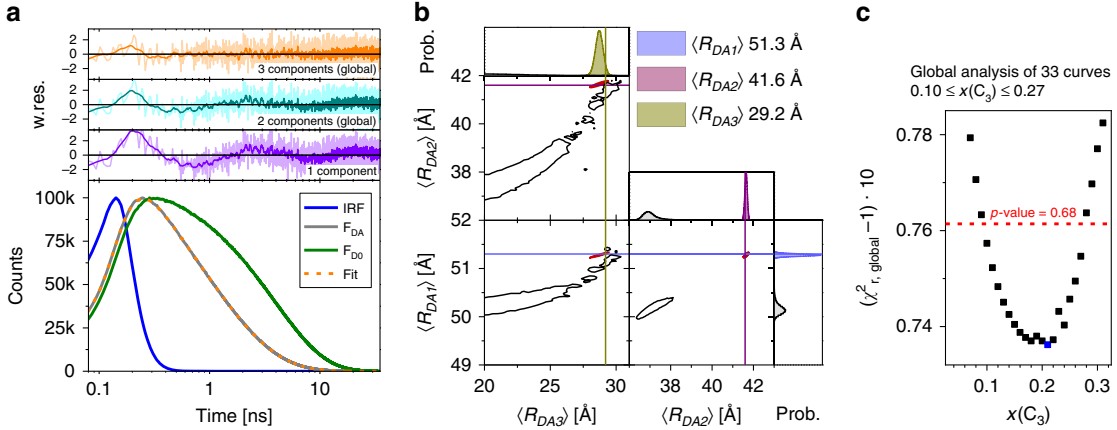

**Fig. 4 Uncovering the third conformational state by eTCSPC.** The eTCSPC measurements of all FRET variants are analyzed by superposition of normal distributed DA-distances (due to the coupling of dyes with long and flexible linkers, details see Supplementary Methods). **a** Experimental fluorescence decay of the variant S36pAcF/P86C ($F_{DA}$, gray, over laid in orange with best fit), the corresponding DOnly-reference sample ($F_{D0}$, green), and instrumental response function (IRF, blue). At the top, the weighted residuals (w.res.) of different analyses by models with DA-distance distributions composed of one (violet), two (cyan), and three (orange) normal distributions are shown in corresponding colors. In the 2- and 3-component analysis, all FRET-measurements are jointly analyzed (global), and the species fractions of the states are shared among all 33 datasets. For the 1-component model of the variant S36pAcF/P86C, we find a mean DA-distance of 45.7 Å with a width of 17.6 Å. The analysis results of the 3-component model are summarized in Supplementary Table 2d–g. **b** Uncertainties, $\Delta R_{DA}(k_{FRET})$, of the 3-component analysis for the variant S36pAcF/P86C. Other contributions to the total uncertainty, $\Delta R_{DA}$, i.e. $\kappa^2$, are not shown for clarity. To the sides, the marginalized (projected) histograms of the sampled model parameters are shown (black: individual fit; red: global fractions). The lines highlight the most likely combination of distances. **c** Uncertainty estimation of the species fraction of the third (minor) state, $x(C_3)$, for the three-component analysis. The fraction $x(C_3)$ was varied from 0–0.32 followed by a subsequent minimization of all other model parameters. This yields the global, reduced $\chi^2_{r,global}$ of all 33 FRET eTCSPC measurements in dependence of $x(C_3)$. This curve has a minimum at $x(C_3) = 0.21$ ($\chi^2_{r,min} = 1.074$). Points above the red line ($\chi^2_r = 1.0761$, $p$-value = 0.68) are significantly worse than the best analysis result as judged by an $F$-test (225 parameters, 100,000 channels).

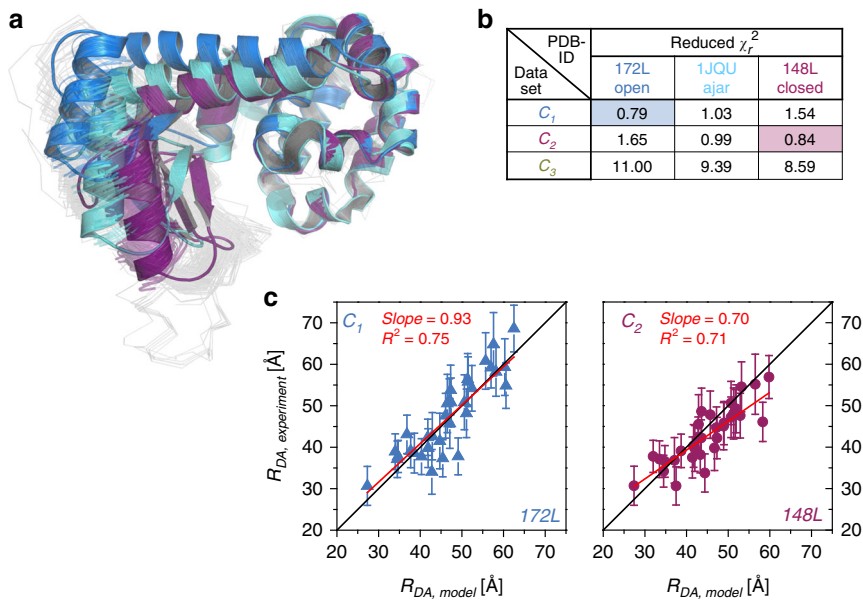

**Fig. 5 PDB screening. a** Overlay of the PDB structures used for screening. Blue, light blue and violet cartoons show the cluster representative of the open (172L), ajar (1JQU), and closed (148L) conformation of T4L. **b** Reduced $\chi^2_r$ for each distance set compared to the expected distances from the selected cluster representative. **c** The experimental distances $R_{DA,experimental}$ of the $C_1$ and $C_2$ dataset are plotted against the model distances $R_{DA,Model}$ from the best PDB structure representative and fitted linearly (red lines). The black lines show a 1:1-relationship. Error bars shown are determined from the support plane analysis (95% confidence interval) shown in Fig. 4 and described in the text.

states[53]. The FRET-experiment with the variant S44pAcF/I150C shows an increase in the population of $C_3$ at pH 2 (Fig. 6b), and the analogous DEER experiment at pH 3 shows a remarkably similar redistribution of interspin distances. Compared to physiological pH conditions (Fig. 6a, dashed trace), these distances exhibit a shortening that is consistent with the $C_3$ state, thus validating our conclusion that EPR and FRET do show the excited state $C_3$.

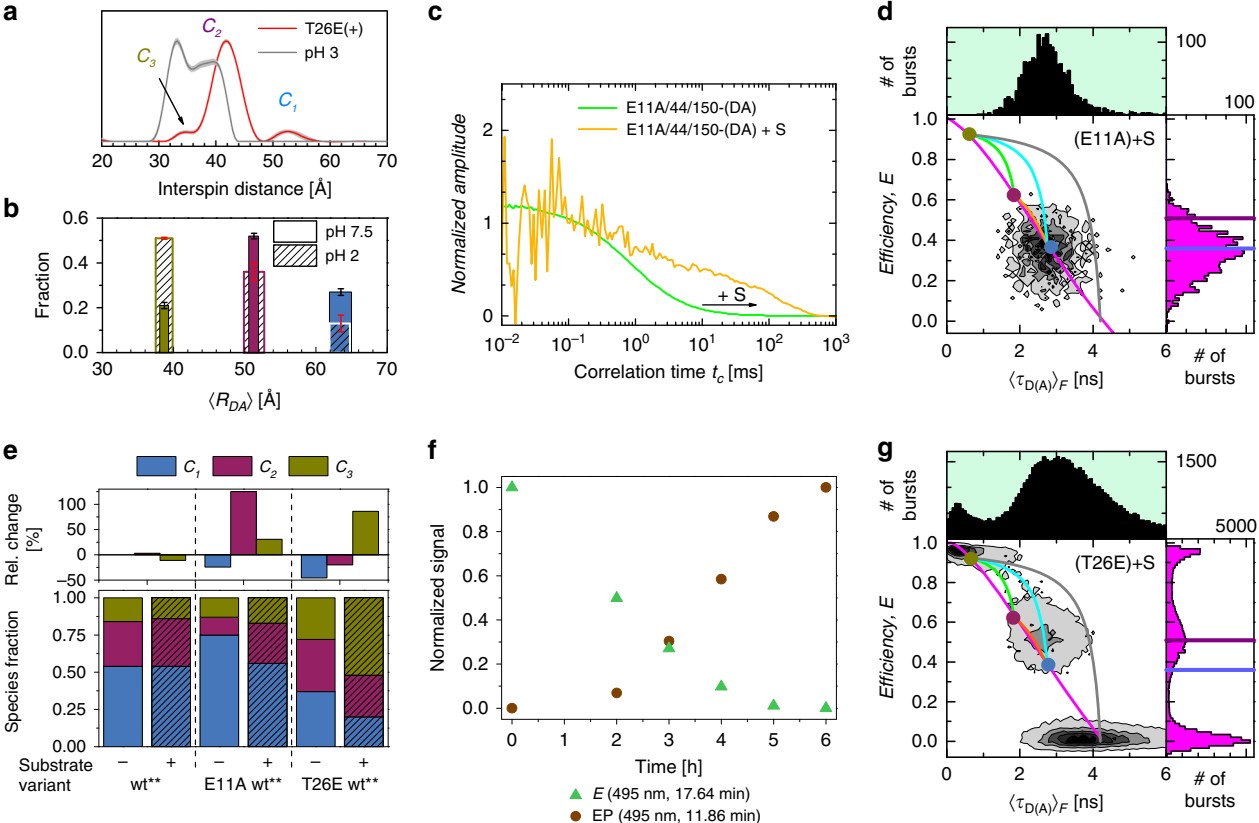

**Fig. 6 Functional states of T4L. a** Interspin distribution of T26E(+)-44R1/150R1 and 44R1/150R1 at pH 3.0 from DEER experiments. Gray area surrounding the distribution corresponds to error estimates. **b** Population fraction of S44pAcF/I150C from eTCSPC at pH 7.0 (solid colors) and at pH 2.0 (dashed rectangles). The width of the bar represents the uncertainty in distance (filled bars: pH 7.5, hatched bars: pH 2). The error bars represent the statistical uncertainty in the amplitudes (black: pH 7.5, red: pH 2). $C_3$ increases in population at low pH. Similar effect is shown in EPR DEER experiments (Supplementary Fig. 5d) error estimates are 95% confidence intervals as determined by support plane analysis (Supplementary Methods). **c, d** The effects of the substrate on E11A/S44C/I150C: **c** Overlay of normalized sCCF of E11A/S44C/I150C with and without substrate. Consistent with the larger rotational correlation, we observe a shift of $t_{diff}$ towards longer times for the variant E11A/S44C/I150C when incubated with the substrate; **d** MFD-histograms for the variant E11A/S44C/I150C with substrate. Upon addition of the substrate, we observe a shift towards lower $E$ values. **e** Species fractions of the variants S44pAcF/I150C (wt**), E11A/S44C/I150C (E11A wt**), and T26E/S44pAcF/I150C (T26E wt**) used to mimic free enzyme (E), enzyme-substrate complex (ES), and enzyme-product bound state (EP) without (−) and with (+) peptidoglycan. At the top, the relative change in fractions upon addition of peptidoglycan is shown. **f, g** Effects of the substrate on the variant T26E/S44pAcF/I150C: **f** Reverse-phase HPLC is used to monitor the adduct formation of the labeled T4L with peptidoglycan via plotting of the normalized signal at 17.64 min (E, green) and 11.86 min (EP, brown) after background subtraction; **g** MFD-histograms for the variant T26E/S44pAcF/I150C with substrate. An accumulation of the high FRET state is observed. The color code and FRET-lines for 2D and 1D MFD histograms (**d**) and **g** are as in Fig. 2.

**Trapped reaction states of T4L.** To mimic functional enzymatic states, we mutated the residues E11 and T26 at the active site using the backbone of the S44pAcF/I150C variant, also named wt**[14,39,54]. We use wt** because of the advantage in clearly resolving all three conformations of the free enzyme (E) by FRET. These mutations help identifying the role of $C_3$ during enzyme catalysis: E11A, which inactivates T4L, causes the enzyme to bind its substrate S (peptidoglycan from *Micrococcus luteus*) while obviating the expected hydrolysis reaction[54]. Thus, in the presence of excess substrate, this mutation mimics the enzyme-substrate complex (ES). We monitor the effect of the substrate binding for the E11A mutation by FCS and compare the characteristic translational diffusion times, $t_{diff}$, in both the absence and presence of substrate. While $t_{diff}$ is small (0.54 ms, Fig. 6c, green curve) without the substrate, it increases by several orders of magnitude when the large substrate is introduced (Fig. 6c, yellow curve). Moreover, the shift towards the larger donor anisotropy values upon incubation with substrate also provides additional evidence for substrate binding without cleavage (Supplementary Fig. 6e).

Sub-ensemble TCSPC analysis of the DA-subpopulation of the ES state (E11A/wt** in the presence of substrate, Fig. 6d, Supplementary Fig. 7a–d, Supplementary Note 5) reveals an increase of 125% in the population corresponding to $C_2$ compared to the free enzyme state $E$, with a concomitant reduction of $C_1$. In contrast, no effect of substrate binding for wt**-(DA) is observed because ES is not trapped (Fig. 6e).

Although the variant T26E cleaves the substrate, the formation of a covalent adduct (PDB ID 148L) prevents a release of the formed product[14]. Therefore, we use this intermediate adduct to mimic the product-bound enzyme state (EP). To confirm the adduct formation under our measurement conditions, we monitor the adduct formation of labeled T4L (T26E/wt** variant) by HPLC (Fig. 6f). T4L without substrate (E) elutes at ~18 min. After incubation with the substrate, the peak of E drops, and a new elution peak at ~12 min is detected with increasing incubation time (Fig. 6f, Supplementary Fig. 8), indicative of the adduct form of T4L (EP). Both ensemble (Fig. 6e) and sm MFD-measurements (Fig. 6g, Supplementary Fig. 7e–h) show a significant increase of the relative fraction of the $C_3$ state, an effect

also observed in the EPR measurements (Fig. 6a). In the T26E variant, the accumulation of the $C_3$ state is connected to the inability of this variant to release a part of the product[14]. We conclude that the new excited conformational state must be involved in this step.

## Discussion

In the following, we present the experimental evidence for the $C_3$ state and its structural properties. To corroborate the existence of $C_3$, we discuss our experiments in four aspects: (1) the kinetic behavior in sm-experiments, (2) the error statistics of data analysis, (3) the structural validation of the obtained FRET parameters, and (4) the effect of specific mutations.

Aspect 1: Kinetic behavior. Considering 18 out of 33 variants with FRET-pairs, the sm-experiments directly show the presence of an additional DA-subpopulation in the MFD-histograms, which differs significantly from $C_1$ and $C_2$ (Figs. 2, 3, Supplementary Fig. 2). This DA-subpopulation is either populated or depopulated with specific mutations that alter the overall catalytic activity of T4L. Moreover, our global fluctuation analysis recovers at least two relaxation times that are shared throughout all studied variants (Fig. 3, Supplementary Fig. 9). Applying kinetic theory, two relaxation times indicate at least three states in equilibrium, which are reproduced by Brownian dynamics simulations (Supplementary Fig. 4).

Aspect 2: Error statistics. Key to the analysis and determination of $C_3$ by ensemble fluorescence decays is the use of global analysis of all 33 variants, which reduces the number of free parameters, increases fitting quality (Supplementary Methods), and gives a consistent description with sm-experiments. Moreover, assuming that CTsD and NTsD are rigid, there are six independent degrees of freedom in the system, which we significantly oversample by measuring 33 variants.

Aspect 3: Structural validation. In contrast to our 3-component model, the global analysis of eTCSPC data using a 2-component model yields two distance sets, which cannot describe the expected interdye distances of the known conformers ($C_1$ (172L) and $C_2$ (148L)). Furthermore, for the 2-component model, we do not observe the expected linear correlation between the modeled and experimental interdye distances, as shown in Supplementary Fig. 5b, c.

Aspect 4: Specific mutations. The final point to corroborate the existence of $C_3$ are the results of a few specific mutations. The variant Q69pAcF/P86C is especially informative, as the donor is placed in the middle of helix c (Orange Fig. 1a), which connects both domains, while the acceptor is located in the middle of helix d, which is part of the CTsD (Brown Fig. 1a). According to FPS, the interdye distances for $C_1$ (172L) and $C_2$ (148L) states are hardly distinguishable by FRET, $\langle R_{DA} \rangle$ of 34 and 35 Å, respectively. Assuming that both domains preserve their secondary structure, the compaction of T4L in $C_3$ can only proceed by kinking the helix c. This conclusion is consistent with previous studies that identify V75 as the subdomain boundary and critical in protein stability of the pseudo-wild-type construct wt* with a boundary for the local stability to unfolding around residue N68[13,55]. Given the location of the dyes and the extension of the dye linkers, expected dye orientations will lead to an increase of the interdye distance for $C_3$, i.e., a greater interdye distance is expected for $C_3$ compared to the experimental distances for $C_1$ (39 Å) and $C_2$ (37 Å). The additional observed distance of 52.4 Å agrees with this hypothesis (Supplementary Table 2f, Fig. 2c).

An additional inactive variant (R137E)[56,57], which disturbs the salt bridge between residues 22 and 137, reduces the population of $C_3$ by ~50% (Supplementary Fig. 10d, Supplementary Table 2h), a

phenotype also observed for the inactive variant E11A/S44C/I150C.

In conclusion, the existence of $C_3$ demonstrates a greater level of complexity of the domain motions of T4L than a single hinge-bending motion, which is in agreement with recent indirect observations[20,24,29]. The complex exchange dynamics between the conformations with relaxation times of 4 and 230 μs and the small population of $C_3$ may explain the difficulties of other experimental biophysical methods and MD simulations in identifying this exchange, and some heterogeneity in interspin distances observed in previous studies for similar conditions[49].

Relating conformational states and enzyme function. A three-step process characterizes the T4L hydrolysis of peptidoglycans. First, the glycosidic bond of the substrate (S) is protonated by E11 followed by the simultaneous nucleophilic attack of water molecules, which are hydrogen-bonded to residues D20 and T26, on the C-1 carbon of S. As a result, the covalent adduct (ES) is observed in PDB ID 148L[14]. Second, the proton is presumably returned from D20 to E11 via solvent transfer. The third and final step is the product release from the active site to regenerate the enzyme to the original state.

In view of the structural dynamics and to link T4L's functional cycle to our three observed conformations, we use an extended MMm (eMMm) scheme as suggested by Kou et al.[30] (Fig. 7a).

Here, this model considers a sequential succession of three steps to go from a free enzyme via a product bound state back to the relaxed free enzyme state. The substrate S binds reversibly to the enzyme E to form an enzyme-substrate complex ES and is converted to the product P, resulting in an EP complex with the product still bound to the enzyme. A transition of E to an excited state E* then releases P from the complex, followed by a relaxation of the free enzyme E* to E.

Next, we consider our results in the light of the eMMm framework. First, by using the S44/I150C backbone and two key functional T4L variants (E11A, T26E), we create the relationship between the conformational ($C_1$, $C_2$, and $C_3$) and the above-described reaction states (E, ES, and EP) for purposes of elucidating the functional role of E* (Fig. 7). We observed a significant difference of the populations of the conformers (Fig. 6) between the three reaction states.

To connect conformational equilibria with catalysis, we monitor the relative changes in species fractions observed across the functional variants in both the absence and presence of substrate (upper panel of Fig. 6e) by generating a $3 \times 3$ state matrix (Fig. 7b). As indicated in the matrix, specific conformational states are favored in each enzyme reaction state (Fig. 7c,d). For this representation, we use the relative population changes as compared to the wt** to monitor the conformational populations of the different enzyme states.

In the free enzyme state E, the open conformation $C_1$ is mostly populated to enable substrate binding, which initiates the catalytic cycle through the formation of ES. Through this cycle, the closed conformation $C_2$ now becomes the most abundant conformation[14]. In this conformation, the glycosidic bond can be cleaved such that $C_2$ connects both ES and EP. In our studies, we determine that the product release may occur in the compact conformation $C_3$, a population that is greatly increased in EP. Thus, $C_3$ links EP and E so that the original enzyme E is regenerated from EP, which closes the enzymatic cycle. Consequently, the compact state $C_3$ now corresponds to the excited conformational state E* (Fig. 7a–d), the function of which is to disperse the product[18,29] (Fig. 7b, d). These series of events show a sequential closing from the most open conformation $C_1$ to the most compact form $C_3$ along two coordinates: the reaction state and the conformational state.

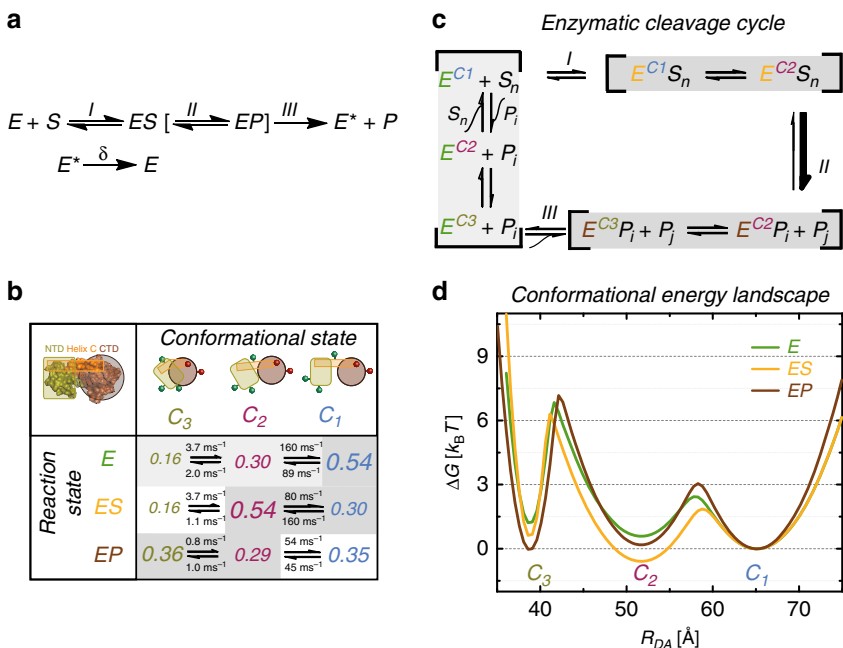

**Fig. 7 Energy landscape of T4L. a** Extended Michaelis–Menten scheme. **b** T4L interconverts between three major $C_1$, $C_2$, and $C_3$ conformational states. The population fractions of $C_1$, $C_2$, and $C_3$ are normalized to the variant S44pAcF/I150C in the absence of substrate using the relative changes in Fig. 6e to correct for the influence of specific mutations in that absence. The different font sizes represent the species fractions $x_i$ for each conformer according to Supplementary Table 2h and satisfy $x_1 + x_2 + x_3 = 1$. The three enzyme states are monitored via the following three enzyme variants: (i) the free enzyme state E via S44pAcF/I150C; (ii) the enzyme-substrate state ES via the inactive E11A/S44C/I150C with bound substrate; and (iii) the enzyme product state EP via the product adduct with T26E/S44pAcF/I150C after substrate cleavage. The reaction rate constants are calculated according to the process detailed in Supplementary Note 3, the confidence intervals of which are shown in Supplementary Table 4c. **c** The peptidoglycan chain with $n$ subunits ($S_n$) is cleaved into two products ($P_i$ and $P_j$ with $n = i + j$) by T4L, both of which can be further processed by T4L until only the dimer of N-acetylglucosamine and N-acetylmuramic acid remains. The gray shaded steps indicate the conformational/reaction states observed. **d** Relative Gibbs free energy landscapes are calculated using $\Delta G^0 = -k_\text{B} T \ln\left(\frac{k_{ij}}{k_{ji}}\right)$, where $k_B$ is the Boltzmann constant and $T$ is the temperature; $k_{ij}$ are the reaction rate constants between states $C_i$ and $C_j$ for the data presented in (**a**). The activation energies are calculated according to $\Delta G^{0\dagger} = -k_\text{B} T \ln\left(\frac{k_{ij}}{k_0}\right)$ assuming $k_0 = 10_3$ ms$^{-1}$ as an arbitrary constant. The distributions consider $C_1$, $C_2$, and $C_3$ to follow a Gaussian distribution as a function of the interdye distance $R_{DA}$. The Gaussian widths ($\sigma_r$) are adjusted to satisfy the energy differences and calculated activation energies. Each energy landscape is independently normalized to $C_1$.

Most strikingly, even under saturating conditions, which favor the ES and EP states, the enzyme was observed to remain in dynamic equilibrium between the conformational states, rather than transforming entirely into a single conformational state.

To visualize the relative energetic changes of the enzyme during the various steps of the catalytic cycle, we use the species fractions and reaction rate constants to compute the relative energy landscape based on the Arrhenius equation (Fig. 7d, Supplementary Fig. 11). We observe a sequential closing of the enzyme to populate $C_3$. This is consistent with a ratchet model for providing directionality on the reaction[58–60] beyond the directionality introduced by the excess of S. This also corroborates with our Monte Carlo simulations (Fig. 3) that are incompatible with the unlikely off pathway from $C_1$ to $C_2$ through $C_3$ due to steric hindrances and with the fast hinge bending motion (4 μs) expected from structural models (PDB ID 172L and 148L).

All our evidence suggests that the conformational state $C_3$, which appears to be more compact than any other structure known of T4L, is compulsory after rather than before catalytic cleavage. Thus, the compact nature of this structure suggests a functional role that is related to product release via an excited state E* (Fig. 7a). This mechanism can be an evolutionary advantage when directionality is required for function. On the contrary, considering a system with only two conformational states and without an active cleaning mechanism, the stochastic dissociation of the product can become rate-limiting given a high affinity of the product to the enzyme in the EP state. Indeed, a

large surfeit of substrate is always characteristic of the in vivo conditions for T4L. Thus, the use of a three-state system to decouple the substrate access and product release can mitigate the occurrence of substrate inhibition in a two-state system when the route to the active site is clogged by excess substrate concentrations[61].

In summary, we studied 33 distinct FRET-pairs to effectively oversample the anticipated simple hinge-bending motion of T4L. Due to the high precision, we identified three substrate-dependent fluorescence states that are in fast kinetic exchange. Inverting the positions of the dyes, e.g. S44pAcF/I150C-DA vs. S44pAcF/I150C-AD (Fig. 3b), rules out specific interactions of the dyes with T4L.

Functional variants change the relative populations of the fluorescence states that are determined in a substrate-dependent manner (Fig. 6). Given successive compaction via three conformational states ($C_1$, $C_2$, and $C_3$), and three reaction states (E, ES, and EP), we considered an eMM reaction scheme (Fig. 7a, c) to provide a meaningful description of the data. Mutagenesis and stability studies indicate the stability of CTsD and a flexibility of the NTsD[4,13,40–42,62], which may be a necessary principle for the construction of enzymes undergoing conformational changes during catalysis. The combination of known structural models and fluorescence data is used to create a proposed novel structural state in the catalytic cycle of T4L involving a rearrangement of the reactive NTsD with respect to the CTsD, which is deemed consistent with the eMMm for enzyme kinetics. For a complete

structural insight, we are now using the obtained FRET-restraints to present a potential model of $C_3$, which is left for a future report.

We anticipate that the presented integrative approach combining the fluorescence spectroscopic toolkit and computational information can accelerate the development of dynamic structural biology[63] by resolving the behaviors of long- and short-lived excited states for purposes of characterizing their functional relevance. This approach is highly relevant as we move towards understanding biomolecular dynamics in situ, where "invisible" molecular effects (i.e., ionic, viscous, and crowding effects[64]) have the potential to modulate weak interactions with important repercussions in biological systems. The elucidation of excited conformational states is necessary for a thorough in-depth understanding of the mechanisms of enzymes. Thus, a comprehensive description of a dynamic molecular system contains intertwined kinetic and structural information, which is often difficult to obtain by traditional methods. Such information can be archived in the data bank PDB-dev[65] so that excited conformational states gain the urgently needed visibility.

## Methods

**Sample preparation**. T4L cysteine and amber (TAG) mutants are generated via site-directed mutagenesis in the pseudo-wild-type construct containing the mutations C54T and C97A (wt*), which was subsequently cloned into the pET11a vector (Life Technologies Corp.)[66-68]. All primer sequences are listed in Supplementary Table 6. The plasmid containing the gene with the desired mutant was co-transformed with pEVOL[66] into BL21(DE3) E. coli (Life Technologies Corp.) and plated onto LB-agar plates supplemented with the respective antibiotics, ampicillin and chloramphenicol. A single colony was inoculated into 100 mL of LB medium containing the above-mentioned antibiotics and grown overnight at 37 °C in a shaking incubator. 50 mL of the overnight culture are used to inoculate 1 L of LB medium supplemented with the respective antibiotics and 0.4 g/L of pAcF (SynChem) and grown at 37 °C until an $OD_{600}$ of 0.5 was reached. The protein production was induced for 6 h by addition of 1 mM IPTG and 4 g/L of arabinose.

Cells are harvested, lysed in 50 mM HEPES, 1 mM EDTA, and 5 mM DTT at pH 7.5 and purified using a monoS 5/5 column (GE Healthcare) with an eluting gradient from 0 to 1 M NaCl according to standard procedures. High-molecular weight impurities are removed by passing the eluted protein through a 30 kDa Amicon concentrator (Millipore), followed by subsequent concentration and buffer exchange to 50 mM PB, 150 mM NaCl pH 7.5 of the protein flow through with a 10 kDa Amicon concentrator. For the double cysteine mutant containing E11A, the temperature was reduced to 20 °C after induction and the cells are grown additional 20 h to increase the fraction of soluble protein. This mutant was produced and purified as described above, except that only ampicillin for selection and IPTG for induction are needed.

Site-specific labeling of T4L was accomplished using orthogonal chemistry. To probe T4L structure by FRET studies (Fig. 2a) we labeled the Keto group of the *p*-acetyl-L-phenylalanine (pAcF) amino acid at the N-terminal subdomain, hydroxylamine linker chemistry was used for the donor dye Alexa488 (Life Technologies Corp.). Cysteine mutants were labeled via a thiol reaction with maleimide linkers of the Alexa647 acceptor dye.

In one exceptional case of the sample S44pAcF/I150C-AD, the labeling was reversed in order to test the reproducibility of the filtered FCS (Fig. 3b) and FRET measurements with different dyes. Acceptor dye Alexa647 with hydroxylamine linker was used to label the pAcF and the Alexa488 donor dye with maleimide linker was used to label the cysteine residue of the mutant S44pAcF/I150C-AD.

For spin labeling, the S44C/I150C double mutant was diluted to a final concentration of ~10 μM in labeling buffer (50 mM MOPS, 25 mM NaCl, pH 6.8) and a 10-fold molar excess of a methanthiosulfonate nitroxide (MTSSL) was added overnight[68]. Next day, excess spin reagent was removed using a desalting column (HiPrep 26/10, GE Healthcare) according to the manufacturer's instructions and concentrated with a 15 kDa Amicon concentrator (Millipore).

Binding of labeled T4L mutants to peptidoglycan from *Micrococcus luteus* (Sigma-Aldrich) was monitored by reverse phase chromatography using a C-18 column out of ODS-A material (4 × 150 mm, 300 Å) (YMC Europe GmbH, Dinslaken, Germany). The labeled protein (1 μM) was incubated with the substrate at 1 mg/mL in PBS. At various points of the reaction, 25 μL of mixed sample injected and further eluted with a gradient from 0 to 80% acetonitrile containing 0.01% trifluoroacetic acid for 25 min at a flow rate of 0.5 ml/min. The labeled complex elution was monitored by absorbance at 495 nm.

**Single-molecule experiments**. For single-molecule measurements with multi-parameter fluorescence detection, we added 40 μM TROLOX to the measurement buffer to minimize the acceptor blinking and 1 μM unlabeled T4L to prevent any adsorption to the cover glass. A custom-built confocal microscope with a dead time-free detection scheme using 8 detectors (four green (τ-SPAD, PicoQuant, Germany) and four red channels (APD SPCM-AQR-14, Perkin Elmer, Germany)) was used for MFD and fFCS measurements. A time-correlated single photon counting (TCSPC) module with 8 synchronized input channels (HydraHarp 400, PicoQuant, Germany) was used to register the detected photon counts in the Time-Tagged Time-Resolved (TTTR) mode. For more details on TTTR please read[69]. The data was analyzed by established MFD procedures[31,33,43] and software, a more detailed description is given in Supplementary Methods. Exemplary data analysis is shown in Supplementary Fig. 3 and Supplementary Note 1, MFD-histograms of all measurements are collected in Supplementary Fig. 2 and Supplementary Fig. 10.

**Filtered FCS**. Filtered FCS (fFCS) is a derivative of fluorescence correlation spectroscopy (FCS). In fFCS, the information on the fluorescent species contained in the time- and polarization- resolved fluorescence decays (exemplary shown in Supplementary Fig. 9a, c) was used to amplify the transitions between the species of interest[32,34,45]. For this, we arbitrarily constructed species-selective filters (exemplary shown in Supplementary Fig. 9b, d) based on the major and minor population in the smFRET experiment and calculated species-selective auto-(sACF) and cross-correlation functions (sCCF). The in total four curves (two sACF's and two sCCF's) are analyzed jointly using established fitting models (Supplementary Methods Equations (15–17))[32,34,45]. For more details see Supplementary Methods.

**Fluorescence decay analysis**. Fluorescence decays of all samples and single-labeled reference samples are collected on either an IBH-5000U (IBH, Scotland) or a Fluotime 200 (Picoquant, Germany) system. We collected high-precision fluorescence decays histograms with 30 million photons to precisely determine the FRET parameters of limiting states together with their corresponding structural properties. eTCSPC has the advantage of better photon statistics, polarization-free measurements due to magic-angle detection, a keenly evolved instrumental response function (IRF), low background fluorescence, and the absence of photobleaching at low excitation powers. As the fluorophores are coupled via long and flexible linkers, this resulted in a DA-distance distribution even for single protein conformations. For our data analysis, we assumed that the dyes rotate quickly ($\kappa^2 = 2/3$) and diffuse slowly compared to the fluorescence lifetime (~ns)[38]. We validated the assumption of fast rotating dyes by time-resolved anisotropy measurements (Supplementary Table 3a–c). Moreover, we interpret the broadening of the DA-distance distributions, beyond what is expected from flexible linkers, as evidence for conformational heterogeneity of the host molecule. To dissect the donor-quenching by FRET (i.e., FRET-induced donor decay), we jointly analyzed the DA- and DOnly-dataset, where the donor fluorescence lifetime distribution was shared with the DA-dataset (Supplementary Methods, Supplementary Equations (20, 24-S27))[47]. We compared three different fit models (Supplementary Table 5). The results are summarized in Supplementary Table 2. We estimated the statistical uncertainty of the model parameters by making use of the known shot noise of the fluorescence decays. We randomly sampled the model parameters by a Markov chain Monte Carlo (MCMC) method to estimate their uncertainties for a single dataset (Supplementary Methods)[70]. The applied joint, global fit significantly reduced the overall dimensionality of the analysis but still left too many degrees of freedom (Supplementary Methods) for an exhaustive sampling by MCMC. Hence, we applied a support plane analysis to estimate the model parameter uncertainties, in which we systematically varied $x_3$ while minimizing all other parameters.

**Electron paramagnetic resonance**. For double electron electron resonance (DEER) measurements of doubly spin labeled proteins, ~200 μM spin-labeled T4L containing 20% glycerol (v/v) was placed in a quartz capillary (1.5 mm i.d. × 1.8 mm o.d.; VitroCom) and then flash-frozen in liquid nitrogen. Sample temperature was maintained at 80 K. The four-pulse DEER experiment was conducted on a Bruker Elexys 580 spectrometer fitted with an MS-2 split ring resonator. Pulses of 8 ns (π/2) and 16 ns (π) are amplified with a TWT amplifier (Applied Engineering Systems). Pump frequency was set at the maximum of the central resonance, and the observe frequency was 70 MHz less than the pump frequency. Dipolar data are analyzed by using a custom program, LongDistances[71], written in LabVIEW (National Instruments Co.). Distance distributions are acquired using Tikhonov regularization[51].

**Recovering the reaction network by Brownian dynamics simulations**. To solve the ambiguity in the connectivity of states and kinetics of T4L, i.e. between the two possible analytical solutions of the transition rate matrix (Supplementary Methods Equation (31)), we used Brownian dynamics simulation of single-molecule and fFCS experiments. Simulations of single-molecule measurements are done via Brownian dynamics[72-75]. The spatial intensity distribution of the observation volume was assumed a 3D Gaussian. In contrast to other simulators, freely diffusing molecules in an "open" volume are used. Transition kinetics is modeled by allowing $i \rightarrow j$ transitions. The time that molecules spend in $i$ and $j$ states ($t_i$ and $t_j$,

respectively) are exponentially distributed with

$$P(t_i) = k_i^{-1} exp(-k_i t_i) \text{ and } P(t_j) = k_j^{-1} exp(-k_j t_j). \quad (1)$$

Simulated photon counts are saved in SPC-132 data format (Becker & Hickel GmbH, Berlin, Germany) and treated as experimental data. To quantify the difference between the two possible, simulated models and the experimental data, we calculated the relative $\chi^2$ for the one-dimensional and two-dimensional MFD-histograms (Supplementary Note 3, Supplementary Table 1a, b).

**Simulation of interdye distances and structural modelling.** Accessible contact volume (ACV) simulations and interdye distances. The accessible volume (AV) considers dyes as hard sphere models connected to the protein via flexible linkers (modeled as a flexible cylindrical pipe)[38]. The overall dimension (width and length) of the linker is based on their chemical structures. For Alexa488 the five carbon linker length was set to 20 Å, the width of the linker is 4.5 Å and the dye radii $R_1$ = 5 Å, $R_2$ = 4.5 Å and $R_3$ = 1.5 Å. For Alexa647 the dimensions used are: length = 22 Å, width = 4.5 Å and three dye radii $R_1$ = 11 Å, $R_2$ = 3 Å and $R_3$ = 3.5 Å. Here, the dye distribution was modeled by the accessible contact volume approach (ACV)[9], which is similar to the accessible volume (AV)[38], but defines an area close to the surface as contact volume.

Similar approaches have been introduced before to predict possible positions for EPR and FRET labels[10,36,37]. The dye is assumed to diffuse freely within the AV and its diffusion is hindered close to the surface. The part of AV which is closer than 3 Å from the macromolecular surface is defined to have higher dye density $\rho_{Dye,trapped}$. The spatial density $\rho_{Dye}$ along $R$ is approximated by a step function: $\rho_{Dye} = [\rho_{Dye,free}, R < 3 \text{ Å}; \rho_{Dye,trapped}, R \geq 3 \text{ Å}]$. The $\rho_{Dye,trapped}/\rho_{Dye,free}$ ratio is calculated from the fraction of the trapped dye $x_{Dye,trapped}$ for each labeling position separately: $\rho_{Dye,free}/\rho_{Dye,trapped} = V_{Dye,trapped} \cdot (1 - x_{Dye,trapped})/ (x_{Dye,trapped} \cdot V_{Dye,free})$. For this, the fraction $x_{Dye,trapped}$ was approximated by the ratio of the residual, $r_\infty$, and fundamental anisotropy, $r_0$, determined by the time-resolved anisotropy decay of the directly excited dyes (Supplementary Table 3).

To account for dye linker mobility, we generated a series of ACV's for donor and acceptor dyes attached to T4L placing the dyes at multiple separation distances. For each pair of ACV's, we calculated the distance between dye mean positions ($R_{mp}$)

$$R_{mp} = \left| \left\langle \vec{R}_{D(i)} \right\rangle - \left\langle \vec{R}_{A(j)} \right\rangle \right| = \left| \frac{1}{n} \sum_{i=1}^{n} \vec{R}_{D(i)} - \frac{1}{m} \sum_{j=1}^{m} \vec{R}_{A(j)} \right|, \quad (2)$$

where $\vec{R}_{D(i)}$ and $\vec{R}_{A(i)}$ are all the possible positions that the donor fluorophore and the acceptor fluorophore can take. However, in ensemble TCSPC (eTCSPC) the mean donor–acceptor distance is observed:

$$\langle R_{DA} \rangle = \left| \left\langle \vec{R}_{D(i)} - \vec{R}_{A(j)} \right\rangle \right| = \frac{1}{nm} \sum_{i=1}^{n} \sum_{j=1}^{m} \left| \vec{R}_{D(i)} - \vec{R}_{A(j)} \right|, \quad (3)$$

which can be modeled with the accessible volume calculation.

The relationship between $R_{mp}$ and $\langle R_{DA} \rangle$ can be derived empirically following a third order polynomial from many different simulations. The $\langle R_{DA} \rangle$ is not directly related to the distance between atoms on the backbone (Cα–Cα), except through the use of a structural model.

FRET positioning and Screening (FPS). FPS is done in four steps, and its flow is based on the recent work by Kalinin et al.[10]. In order to do our experimental design using the available PDB structures of T4L with respect to our FRET data, FPS calculates the donor and acceptor accessible volumes for each donor–acceptor labeling scheme. We then compute an error function for each conformational state $C^{(i)}$

$$\chi^2_{r,FPS^{(j)}} = \frac{1}{N} \sum_{i=1}^{N} \frac{\left( \langle R_{DA} \rangle^{(i)}_{experiment} - \langle R_{DA} \rangle^{(i)}_{model \, j} \right)^2}{\left( \Delta R^{(i)}_{DA,tot} \right)^2}, \quad (4)$$

where $N = 33$ is the total number of FRET distances ($\langle R_{DA} \rangle$) and the overall theoretically computed absolute uncertainty $\Delta R^{(i)}_{DA,tot}$ (see next section).

In order to compare the structural models currently available in the PDB to our experimental results, we clustered all PDB models using the RMSD (Root Mean Squared Deviation) of Cα atoms as the similarity measure. Clustering allowed us to sort all PDB models into three distinct groups based on the similarity of their backbone shapes. We found that the structural models of T4L group into open, ajar, and closed clusters (based on the proximity of the CTsD and NTsD) with an intra-cluster RMSD of less than 1.8 Å. Representative structures of these clusters are given by PDB IDs 172L, 1JQU, and 148L for the open, ajar, and closed conformations, respectively (Fig. 5a). This was done using the agglomerative hierarchical complete-linkage clustering of the "fastcluster"[76] software.

In Supplementary Table 4 we provide the complete breakdown of the three clusters. In Supplementary Fig. 6 we display the complete FRET-screening of the three clusters.

**Calculation of uncertainties.** The overall theoretically computed absolute uncertainty $\Delta R^{(i)}_{DA,tot}$ of the average inter-dye distance for the pair $(i)$ is determined by the error-propagation rule:

$$\Delta R^{(i)}_{DA,tot} = \sqrt{\left( \delta R_{dye \, model} \cdot R^{(i)}_{DA} \right)^2 + \left( \delta R_{R_0} \left( r(t)_{dye} \right) \cdot R^{(i)}_{DA} \right)^2 + (\Delta R_{Reference})^2 + \left( \Delta R^{(i)}_{noise-,+} \right)^2} \quad (5)$$

In the following, we describe the computation of the four individual contributions expressed as absolute and relative distance uncertainties, $\Delta R$ and $\delta R$, respectively.

(1) Dye model. The relative distance error $\delta R_{dye \, model}$ usually considers the asymmetry of the AVs for random labeling of two equivalent labeling sites (in general two cysteines) with $\delta R_{dye \, model} \approx 1.5 \%$[47]. However, in this work we labeled T4L specifically (one cysteine, one p-acetylphenylalanine) so that $\Delta R_{dye \, model} = 0$.

(2) Uncertainty of the Förster Radius $R_0$. The relative distance error $\delta R_{R_0}(r(t)_{dye})$ considers the uncertainty of the Förster Radius $R_0$ that is usually dominated by $\kappa^2$ errors related to the mutual orientation of donor and acceptor. At first, we use the experimental anisotropy decays $r(t)_{dye}$ recorded by eTCSPC and MFD and the wobbling in a cone (WIC) model to compute possible distribution of orientational factors, $p(\kappa^2)$[38,77]. As input we determined the residual anisotropies of the donor fluorescence $r_{3,D}$ (Supplementary Table 3a), the directly excited acceptor fluorescence $r_{3,A}$ (Supplementary Table 3b) and the FRET-sensitized acceptor fluorescence $r_{2,A(D)}$ (Supplementary Table 3c). Based on these limits, we computed the distribution of the orientation factor $p(\kappa^2)$ (Supplementary Table 3d) for each FRET pair as described in Sindbert at al.[38]. Next, we compute how $p(\kappa^2)$ affects the experimentally observed interdye distance. Following the approach of Sindbert et al.[38], we can assume that a DA pair is characterized by a single "true" DA distance $R_{DA}$ with $\kappa^2 \neq 2/3$. As we calculate the DA distance assuming $\kappa^2 = 2/3$, we only recover an apparent DA distance, $R_{app}$. Obviously, $R_{app}$ differs from $R_{DA}$, $R_{app}/R_{DA} = \left( \frac{3}{2} \cdot \kappa^2 \right)^{-1/6}$. A distribution of $\kappa^2$ relates for a single $R_{DA}$ to a distribution of apparent $R_{app}$. For each FRET pair the distribution $p(\kappa^2)$ compiled in Supplementary Table 3d is transformed to a distribution of the relative distances $R_{app} / R_{DA} = \xi$. The standard deviation of the distribution $p(\xi)$ is used as a relative approximate for the precision of the distance $R_{DA}$.

$$\delta_{R_{DA},precision}(\kappa^2) \approx (Var[\xi])^{1/2} \quad \text{with } \xi = R_{app}/R_{DA} \quad (6)$$

The expectation value of $p(\xi)$ can be used as an estimate of the accuracy:

$$\delta_{R_{DA},accuracy}(\kappa^2) \approx \langle \xi \rangle - 1 \quad \text{with } \langle \xi \rangle = \int \xi \cdot p(\xi) d\xi \quad (7)$$

For the WIC model with the given residual anisotropies, the precision $\delta_{R_{DA},Precision}(\kappa^2)$ dominates the relative uncertainty. Estimates for $\delta_{R_{DA},accuracy}(\kappa^2)$ are very close to one. Hence, we estimate the overall uncertainty attributed to $\kappa^2$ by

$$\delta R_{R_0} \approx \delta_{R_{DA}}(\kappa^2) = \delta_{R_{DA},precision}(\kappa^2) = \left( \int (\xi - \langle \xi \rangle)^2 \cdot p(\xi) d\xi^{1/2} \right) \quad (8)$$

(3) Uncertainty of the D-only reference. The absolute uncertainty of the Donor-only reference $\Delta R_{reference}$ considers the discrepancy between the photophysical properties of the donor in the FRET sample and the properties determined from the donor-only sample. This discrepancy is typically caused by unspecific labeling of the biomolecule and thus unknown fraction of donor–acceptor molecules with respect to acceptor-donor molecules (see also Peulen et al.[47], Fig. 12). In this work, $\Delta R_{reference}$ was set to zero, since specific labeling was used and the donor position is known exactly and accurate donor only sample was measured (Supplementary Table 2a).

(4) Statistical uncertainty (error of the fit). The state-specific asymmetric absolute statistical uncertainty $\Delta \left\langle R^{(i)}_{noise-,+} \right\rangle$ is caused by the shot noise of the lifetime measurements (listed in Supplementary Table 2g). $\Delta \left\langle R^{(i)}_{noise-,+} \right\rangle$ is calculated from the spread of obtained distances for the three states of the global fit using the shortest ($\langle R_{noise-} \rangle$) and longest distance ($\langle R_{noise+} \rangle$) below the 1σ-threshold (Supplementary Table 2g). As the distance $\langle R_{DA} \rangle$ of the global fit with the lowest $\chi_r^2$ ($x_3 = 0.18$) is not necessarily the average of $\langle R_{noise-} \rangle$ and $\langle R_{noise+} \rangle$, both $\Delta \left\langle R^{(i)}_{noise-,+} \right\rangle$ and the resulting $\Delta R^{(i)}_{DA,tot}$ are not symmetric:

$$\Delta \left\langle R^{(i)}_{noise-,+} \right\rangle = \left| \left\langle R^{(i)}_{noise-,+} \right\rangle - \left\langle R^{(i)}_{DA} \right\rangle \right|.$$

For example, considering a 1σ-confidence interval, the fraction $x_3$ of $\left\langle R^{(3)}_{DA} \right\rangle$ varies between 0.1 and 0.27 (see Fig. 4c). The corresponding $\left\langle R^{(i)}_{noise-} \right\rangle$ and $\left\langle R^{(i)}_{noise+} \right\rangle$ are the shortest and longest distance below the 1σ-threshold. The minimal $\chi^2_{r,global}$ for this distribution fit model is 1.0736 with the species fraction $x_3 = x_{middle} = 0.18$ with $x_1 = 0.44$ and $x_2 = 0.38$ and the state-specific mean distances $\left\langle R^{(i)}_{DA} \right\rangle$ are listed in the Supplementary Table 2d–f.

**Controls for the FRET analysis.** Since the problems inherent in the use of smFRET studies are connected with complexities related to the labels, we performed ten controls to check for any potential label artifacts. Please refer to the

Supplementary Information for additional data and extended controls (Supplementary Note 6).

1. The labeling does not alter enzyme function with the labeled T26E variant indicating an expected adduct formation (Fig. 6, Supplementary Fig. 8).
2. Local quenching of the donor dye is considered when calculating distances and cross correlations (Supplementary Table 2).
3. The triplet state quenchers do not affect the observed relaxation times and amplitudes on the species cross-correlation (Supplementary Fig. 12).
4. The acceptor cis-trans isomerization does not contribute to the signal on the species correlation analysis (Supplementary Fig. 12).
5. The $\kappa^2$ distributions indicated the validity of our assumption of $\kappa^2 = 2/3$. Supplementary Table 3 summarizes the residual anisotropies ($r_\infty$) of D—donor, A—acceptor and A(D) by the FRET-sensitized emission of acceptor used for calculating $\kappa^2$ distributions[31].
6. The existence and the population fraction of the new conformational state $C_3$ with a confidence rate of 68% between 10 and 27% is consistent across our library of 33 variants with $x_3 = 21\%$. The variation of the experimental uncertainty is consistent with the determination in the literature that mutations slightly affect the conformational stability of T4L, which was measured in chemical denaturation experiments[62]. We thus attribute this variability of the species fractions to mutation effects.
7. All 33 variants provide a consistent view of the T4L conformational states, in which we determined after X-ray crystallography a consistency with the two limiting structures determined by T4L without outliers (Fig. 5, Supplementary Table 2d–f).
8. We oversample the FRET restraints to reduce the uncertainty introduced from each point mutation (Fig. 5, Supplementary Table 2d–f).
9. The thermodynamic stability and proper folding of our variants are verified by chemical denaturation with urea and by measuring CD spectra for both unlabeled and labeled T4L.
10. We fit time resolved fluorescence decays with various models to provide a consistent view of the conformational space (Supplementary Methods).

**Reporting summary**. Further information on research design is available in the Nature Research Reporting Summary linked to this article.

## Data availability

The source data for all variants and all used techniques and screening results for the PDB structures were uploaded to Zenodo under DOI 10.5281/zenodo.3376527 and are described in Supplementary Note 7. The Zenodo archive contains subfolders for all eTCSPC FRET data including reference measurements used for derivation of DA distances (eTCSPC_wildtype.zip; e.g. Figures 4, 5, and Supplementary Fig. 5), single-molecule raw data used for filtered FCS and MFD analysis (Single_molecule_wildtype. zip; e.g. Figures 2, 3, Supplementary Figs. 2, 4, and 9), single molecule data of functional variants used for derivation of the kinetic scheme (Single_molecule_functional_variants. zip; e.g. Figure 6, Supplementary Figs. 6, 7, 9, and 11), EPR data (EPR_wildtype.zip; e.g., Figure 6 and Supplementary Fig. 5), and an overview on the FRET screening of PDB structures (FRET_screening_of_PDB_structures.zip). The source data underlying Supplementary Fig. 8c is shown in the Source Data File. Further datasets of processed data and the analysis are available from the corresponding authors on reasonable request.

## Code availability

Most general custom-made computer code is directly available from http://www.mpc. hhu.de/en/software. Additional computer code custom-made for this publication is available upon request from the corresponding authors. In-house programs are used (1) in the confocal multiparameter fluorescence detection experiments, (2) to elucidate the filtered fluorescence correlation spectroscopy curves, and (3) to analyze the fluorescence lifetime measurements. Software for analysis of single-molecule measurements and fluorescence correlation analysis and their simulation is available at http://www.mpc.uni-duesseldorf.de and software for analysis of fluorescence decays can be downloaded from http://www.fret.at/tutorial/chisurf/.

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

## Acknowledgements

We wish to acknowledge the support of Marina Rodnina, Philipp Neudecker, and M. Neal Waxham for their comments and suggestions on the paper, and Daniel Rohrbeck for support on the mutagenesis. We acknowledge the support of Christian A. Hanke for helping in organizing the data deposition at Zenodo. We also wish to acknowledge Evan Brooks for generating several of the T4L variants and in assisting with the DEER sample preparation. This research was supported by the European Research Council through the Advanced Grant 2014 hybridFRET (671208) (to C.A.M.S.), by the NIH (grant R01EY05216 to W.L.H. and grant 1P20GM121342 to HS), NSF CAREER MCB (grant 1749778 to H.S.) and the Jules Stein Professorship Endowment (to W.L.H.). H.S. wishes to acknowledge the support from the Alexander von Humboldt foundation and the Clemson Start-up funds. K.H. wishes to acknowledge the support of the NRW Research School BioStruct and the iGRASPseed International Graduate School of Protein Science & Technology.

## Author contributions

H.S. and K.H. purified and labeled the protein. H.S., T.P., K.H., and D.R measured and analyzed the FRET experiments. T.P., M.D., and H.G. performed structural screening. M.R.F. performed and analyzed EPR experiments. W.H. performed study design and EPR analysis. S.F. developed fluorescence analysis tools. F.K. and R.K. developed fluorescence instrumentation and gave technical support. All authors discussed the results and commented on the paper. H.S., K.H., W.H., and C.A.M.S. wrote the paper. H.G. contributed to the writing of the paper. C.A.M.S. supervised the project.

## Competing interests

The authors declare no competing interests.
