## [Peer Review File · Nature Communications]

Reviewers' comments:

Reviewer #1 (Remarks to the Author):

Single-molecule multi-parameter and time-resolved spectroscopy is a powerful approach for characterizing inhomogeneous and complex chemical rate processes of enzymes. Conformational motions of enzymes at the single-molecule level are often dynamic and stochastic. Obtaining molecular level insights into conformational transition dynamics of enzymes from the inactive state to the active state and/or productive vs non-productive conformational motions are the fundamental problems for understanding enzymatic catalytic and conformational dynamics. Patterned/ordered exponential-to-nonexponential Gaussian-like distributions of the time duration in forming the active state (ES*) through multiple-intermediate conformational states, the bunching feature of conformational motion (bunching effect), and multiple pathways of enzymatic reaction product releasing have been observed and reported in recent years, but the detailed identification of the transient and fluctuating intermediate enzymatic states are still a high challenge for experimental advancement in modern biology. In this manuscript, the authors have demonstrated an elegant study to characterize the single-molecule structure-dynamics-function study of complex T4 lysozyme enzymatic reaction dynamics and mechanism. This is a highly significant and important work from both technical demonstration and scientific knowledge advancement perspectives. This study significantly advanced, technically and conceptually, the protein function-dynamics-function knowledge and approaches that the dynamic structural biology of T4L plays a critical role in an enzymatic reaction function by using a novel and state-of-the-art combining single-molecule and ensemble multi-parameter fluorescence detection, EPR spectroscopy, mutagenesis, and FRET-positioning and screening, which provides a direct characterization of three short-lived conformational states within the conformational landscape of the T4L over the ns-ms. The manuscript is well written, and it should be publishable after a minor revision. Considering the significance of this work, I recommend an urgent publication of this manuscript. I have the following suggestions for the authors to consider in a revision:

(1) In abstract and conclusion of the manuscript, the authors stated that "The present fluorescence spectroscopic toolkit is anticipated to accelerate the development of dynamic structural biology." Single-molecule enzymology has come a long way since the early demonstrations of the single-molecule spectroscopy studies of enzymatic dynamics about two decades ago. The rapid developments of this fundamental protein dynamics field are hand-in-hand with the new developments of single-molecule imaging and spectroscopic technology and methodology, theoretical model analyses, and correlations with biological preparation and characterization of the enzyme protein systems. For a broader readership, it should be beneficial if the authors can elaborate this statement to give more specifics on how the complex and fluctuation conformational states may play critical roles in an enzymatic reaction and how this reported approach provides a novel capability for specifically addressing such important and broadly relevant problems for the modern biology, beyond the enzymatic enzymology as the authors have demonstrated beautifully in the work.

(2) It would be good for the readership to add a paragraph before the conclusion or in the main text to discuss the complexity of the local environment and individual molecule fluctuation dynamics at the enzymatic reaction active site.

(3) It will be beneficial for a broad readership to appreciate the significance of the technical and biological broad impact if the authors may consider adding an explanation in the supporting information on (1) the technical details of the biological enzymatic reaction assay; (2) the photon stamping spectroscopic detection on timing and time resolution: an essential description for a broader readership; and (3) the typical limitations of the complex technology, and some future possible alternative advancement for addressing more complex protein dynamics problems in studying single molecule in living cells or in a crowding molecular environments.

Reviewer #2 (Remarks to the Author):

The authors use an impressive combination of methods to investigate T4 lysozyme (T4L). In particular, they use various single molecule fluorescence based methods and EPR spectroscopy supported by biochemical assays. The findings are novel and they will be of interest to others in the community. In my opinion, they have a large potential to influence thinking in the field.

First the authors find a third new state for T4 lysozyme in addition to the two previously known states. In my view this third state is very convincing in particular as 33 distances were screened. I do not understand, why the authors do not show a model (structure) for this C3 state. If I understand correctly, all data is already in Figure S5 and a structure would be nice to see.

Second, the authors characterize the dynamics and function of this C3 state and its connection to the C1 and C2 states. This is not as convincing as the existence of this third state. I still do not fully understand the line of arguments here, probably answering the following questions would help:

1. Page 8: "In the presented data, the major populations are shifted to the right of the static FRET line" - I cannot see that, in particular in Fig. 2C. How is this justified/quantified? If I understand correctly, this is the main argument for C3 being in dynamic exchange?
2. Species cross correlation function: It remains unclear to me how the pseudo-species are selected and how relaxation times for the real species can be extracted, although the specific amplitudes and their relationships are lost.
3. Review of NCOMMS-18-17509eTCSPC: When the instrument response is subtracted, there are basically no features in the curve. Although the residuals look perfect, I would expect that a multitude of solutions with three relaxation times will fit the data as well. Is it possible to give confidence intervals for the fit parameters?
4. eTCSPC data: If I understand this correctly, there is no example where a single relaxation time is sufficient to describe the data, which worries me. If T4L is trapped in a single (fixed) state, there should be only a single relaxation time (distance)? In addition, how are the three conformations assigned to the three decay times to start with? Is this assignment left completely open to start the global analysis?

Finally, the authors derive a free energy landscape for the three states. As far as I understand, this analysis completely relies on the assumption of a three state system with a linear succession of the states C3,C2 and C1. For the details on enzyme function even a certain model (eMMm) is assumed and all results concerning catalysis are only valid if this model is correct. These are strong assumptions and many other (especially larger) models would also fit the data well. Therefore, this has to be clarified and emphasized throughout the text. In particular:

5. Page 25: I cannot see how to observe a certain sequence of states from the presented data. This is s.th. that has been put into the model. In my opinion here you only get as a result what you already have put in!
6. Page 28: "a detailed enzymatic reaction scheme is developed": No - this is assumed.
7. Kinetic network: I do not understand the argument of "microscopic reversibility of transitions" to discriminate the full cycle. A cycle can also be reversible.
8. I do not understand the argument for the transition C1-C3 being forbidden (Supplement 2.6.2.), maybe this can be clarified, because it is a very important point in the argument for the chosen model. In particular, as fFCS shows at least two transition times faster than 10ms (this could also be three) and why is C3t static?

Minor comments:

1. Page 8: How does MFD compare to NMR relaxation dispersion measurements - I cannot find this information in the Supplementary Information as indicated.
2. Page 25: The authors use the Arrhenius equation with a pre-factor of 10e-03 ms. I have not seen that before, usually 10e-05 ms or less are used (e.g. Joe Howard:"Mechanics of Motor Proteins and the Cytoskeleton"). In addition, I suggest to use the unit sec or Hz.
3. Page 26: I do not understand the word "processivity" in the context of this enzymes. Please

explain.

4. Fig. S2B: I cannot see a "more pronounced tilt towards the C3". What I see is a broadening of the peak, which also could be caused by a tilt towards C2.

Reviewer #3 (Remarks to the Author):

This evaluation refers only to the DEER results of this manuscript

The authors use DEER measurements of one of their many mutants to support the FRET based finding of the existence of three conformations. It is well known that distance distributions derived using Tikhonov regularization often show some shoulders and minor peaks in the distance distributions and this often depends on the regularization parameter chosen for the analysis. Therefore the assignment to different conformations of the protein is highly questionable (can be also originate from the label) unless some additional systematic experiments are carried out to substantiate such assignment. The authors should carry out validations available in the DEERanalysis software to highlight uncertainties, these are usually pretty good in revealing what features in the distribution can be trusted. These, together with the primary DEER data should be added to the SI.

I scanned the literature and there are many papers showing DEER measurements on a variety of double mutants of T4L using different types of nitroxide spin labels and even a few Gd(III) based spin labels. All these usually show one peak in the distance distribution. Particularly narrow distance distributions are obtained with double arm labels (both nitroxide and Gd(III)). In the case of the nitroxide the comparison with the usual R1 label demonstrate well the effect of the label in the broadening. Although, the labeling sites are not exactly the same as in this paper they are not too far away and I would expect to see the different conformations there but they do not even show two resolved conformations. This should be discussed.

I would interpret the DEER data more cautiously as follows: the pH=7.5 has a broad distance distribution which, in addition to the spin label induced broadening (it can actually be estimated from the crystal structures using MMM) reflects the flexibility of the protein and represents a family of conformations. The linked construct restricts the dynamics of the protein leading to a narrower distance distribution. The change of pH leads to a similarly broad distribution as the 7.5 pH sample, with but with a shorter mean distance, indicating that different conformations are present. I am really not convinced that these data show that three well distinct conformations are present in the pH=7.5.

Minor comments

Better use the name MTSSL and not HO-225 for the nitroxide agent used to label the cysteine
Why is the distance distribution cut a little below 3 nm, cutting the dashed distance distribution?
Please show the distance distribution starting at 1.5 nm.

Changes on the manuscript are highlighted in yellow.

Reviewer 1:

Single-molecule multi-parameter and time-resolved spectroscopy is a powerful approach for characterizing inhomogeneous and complex chemical rate processes of enzymes. Conformational motions of enzymes at the single-molecule level are often dynamic and stochastic. Obtaining molecular level insights into conformational transition dynamics of enzymes from the inactive state to the active state and/or productive vs non-productive conformational motions are the fundamental problems for understanding enzymatic catalytic and conformational dynamics. Patterned/ordered exponential-to-nonexponential Gaussian-like distributions of the time duration in forming the active state (ES*) through multiple-intermediate conformational states, the bunching feature of conformational motion (bunching effect), and multiple pathways of enzymatic reaction product releasing have been observed and reported in recent years, but the detailed identification of the transient and fluctuating intermediate enzymatic states are still a high challenge for experimental advancement in modern biology. In this manuscript, the authors have demonstrated an elegant study to characterize the single-molecule structure-dynamics-function study of complex T4 lysozyme enzymatic reaction dynamics and mechanism. This is a **highly significant and important work from both technical demonstration and scientific knowledge advancement perspectives**. This study significantly advanced, technically and conceptually, the protein function-dynamics-function knowledge and approaches that the dynamic structural biology of T4L plays a critical role in an enzymatic reaction function by using a novel and state-of-the-art combining single-molecule and ensemble multi-parameter fluorescence detection, EPR spectroscopy, mutagenesis, and FRET-positioning and screening, which provides a direct characterization of three short-lived conformational states within the conformational landscape of the T4L over the ns-ms. The manuscript is well written, and it should be publishable after a minor revision. Considering the significance of this work, I recommend an urgent publication of this manuscript. I have the following suggestions for the authors to consider in a revision:

We thank the reviewer for identifying the significance of this work and the very positive comments in favor to accelerate publication given the impact of the presented work. We use the recommended suggestions from the reviewers to improve the manuscript, given the word limits, as follows.

(1) In abstract and conclusion of the manuscript, the authors stated that “The present fluorescence spectroscopic toolkit is anticipated to accelerate the development of dynamic structural biology.” Single-molecule enzymology has come a long way since the early demonstrations of the single-molecule spectroscopy studies of enzymatic dynamics about two decades ago. The rapid developments of this fundamental protein dynamics field are hand-in-hand with the new developments of single-molecule imaging and spectroscopic technology and methodology, theoretical model analyses, and correlations with biological preparation and characterization of the enzyme protein systems. For a broader readership, it should be beneficial if the authors can elaborate this statement to give more specifics on how the complex and fluctuation conformational states may play critical roles in an enzymatic reaction and how this reported approach provides a novel capability for specifically addressing such important and broadly relevant problems for the modern biology, beyond the enzymatic enzymology as the authors have demonstrated beautifully in the work.

The abstract is improved as suggested.

“The presented fluorescence spectroscopic toolkit is anticipated to accelerate the development of dynamic structural biology **by identifying transient conformational states that are highly abundant in biology with critical roles in enzymatic reactions.**”

(2) It would be good for the readership to add a paragraph before the conclusion or in the main text to discuss the complexity of the local environment and individual molecule fluctuation dynamics at the enzymatic reaction active site.

We improved the closing paragraph on the Conclusion section to frame the impact of this approach and comment on the influence of local environment in the dynamic fluctuations function

“We anticipate that the presented fluorescence spectroscopic toolkit can accelerate the development of dynamic structural biology by resolving the behaviors of long- and short- lived excited states for purposes of characterizing their functional relevance. The elucidation of these conformational states is necessary for initiating a thorough in-depth understanding of the mechanisms of enzymes. **This approach is highly relevant as we move towards understanding biomolecular dynamics *in situ*, where “invisible” molecular effects (i.e. ionic, viscous, and crowding effects⁵⁹) have the potential to modulate weak interactions with important repercussions in biological systems.**”

(3) It will be beneficial for a broad readership to appreciate the significance of the technical and biological broad impact if the authors may consider adding an explanation in the supporting information on (1) the technical details of the biological enzymatic reaction assay; (2) the photon stamping spectroscopic detection on timing and time resolution: an essential description for a broader readership; and (3) the typical limitations of the complex technology, and some future possible alternative advancement for addressing more complex protein dynamics problems in studying single molecule in living cells or in a crowding molecular environments.

(1) We thank the reviewer for noting that we missed the inclusion of some details on the description of the condition of our measurements. The new version contains these details on the online methods section.

(2) Photon stamping: On the methods section, we added the time resolution (1 ps); other specifics were already present. We also added the reference to the application note of Picoquant GmbH, as suggested literature to obtain a description for broader readership⁶³. (https://www.picoquant.com/images/uploads/page/files/14528/technote_tttr.pdf)

(3) Limitations: We added a paragraph on Supplementary Note 6 (index 9) stating the limitations.

“Limitations on hybrid FRET: Current limitations of hybrid FRET include the introduction of labels via site directed mutagenesis. Not all systems are resistant to extensive mutations and maintain their stability. In terms of the size of molecules under study, the limitations will be determined by the level of resolution that it is desired. Larger molecules would require a larger set of FRET measurements and in some cases the inclusion of fluorescent proteins, which would cause larger uncertainties. Although, nothing limits the ability to do measurements in complex environments (crowder or viscous solutions and even in live cells), one must assure that proper controls are performed. Of particular interest are measurements in living cells, where the labeled molecules have to be introduced without damaging the liability of the cells.”

Moreover, we thank the reviewer for raising this topic. This field is in constant development. Unfortunately, given the context, we feel that to fully address this point, we would require significant more space than given in this manuscript. Also, we consider that that content is better fit in other follow up manuscript that we currently have on the pipeline. We do acknowledge that this is an important aspect of the scientific progress and we are working on making the transfer of knowledge more efficiently. We appreciate the reviewer for this request and the insightful review.

Reviewer 2:

The authors use an impressive combination of methods to investigate T4 lysozyme (T4L). In particular, they use various single molecule fluorescence based methods and EPR spectroscopy supported by biochemical assays. The findings are novel and they will be of interest to others in the community. In my opinion, they have a large potential to influence thinking in the field.

We thank the reviewer for the in-depth review. By addressing the raised concerns, we were able to improve the clarity and content.

First the authors find a third new state for T4 lysozyme in addition to the two previously known states. In my view, this third state is very convincing in particular as 33 distances were screened. I do not understand, why the authors do not show a model (structure) for this C3 state. If I understand correctly, all data is already in Figure S5 and a structure would be nice to see.

As noted by the reviewer, there is a need to have a structural model of this new states (C3). However, modeling a structure assisted by FRET derived distances is not as trivial as one might think. We are in the process of refining a structural model. The methods needed for modeling this state require a thorough development and testing (Peulen T.O. et al J. Phys Chem 121(35), 8211, 2017 PMID: 28709377, [10.1021/acs.jpcc.7b03441](https://doi.org/10.1021/acs.jpcc.7b03441)) and a lengthy description on how we acquired the structure; therefore, it will be unrealistic to fit everything within a single manuscript. Therefore, we are leaving this work for future manuscripts, one of which will be submitted soon. Nonetheless, we assure the reviewer that is one of our most current and pressing priorities as we are establishing all necessary documentation so that FRET-assisted structural models can be deposited in a public database (<https://pdb-dev.wwpdb.org>) using the FRET dictionary (<https://github.com/Fluorescence-Tools>). Therefore, we expect that once the methods have been established, we will submit our structural models to this database.

Second, the authors characterize the dynamics and function of this C3 state and its connection to the C1 and C2 states. This is not as convincing as the existence of this third state. I still do not fully understand the line of arguments here, probably answering the following questions would help:

1. Page 8: "In the presented data, the major populations are shifted to the right of the static FRET line" - I cannot see that, in particular in Fig. 2C. How is this justified/quantified? If I understand correctly, this is the main argument for C3 being in dynamic exchange?

To simplify the observation, we added in Fig 2C dashed lines at the peak of the distribution to observe the location with respect of the FRET-line. This is a qualitative justification.

However, this is not the main argument to justify that C3 is in dynamic equilibrium with the other states. To clarify this, we added a diagram as a Supplemental Figure 1 describing the protocol to identify dynamics.

Shortly, here are the steps.

Do MFD histograms resolve multiple states in the ms timescales?

If the answer is yes, the next question is. Does TCSPC resolves the same states at the nano-second timescales?

If the answer is yes, those states are stable on the ms timescales. Thus, no dynamic averaging is observed. Otherwise, there is dynamic averaging. Then, we use the FRET-lines as guidelines to propose kinetic models, which then are tested later.

If submillisecond dynamic averaging is observed, we use PDA and fFCS to identify the characteristic times or reaction rate constants, if the model permits. PDA is useful for simple

kinetic networks and for dynamics that occur at timescales similar to the diffusion time. On the other hand, fFCS is good in identifying the relaxation times on time scales from the laser repetition rate up to the diffusion time, providing a broader dynamic range.

Generally, the number of relaxation times observed in fFCS correspond to $N-1$ number of states. This is true, because the relaxation times are the eigenvalues of the transition rate matrix in a $N \times N$ kinetic network.

2. Species cross correlation function: It remains unclear to me how the pseudo-species are selected and how relaxation times for the real species can be extracted, although the specific amplitudes and their relationships are lost.

To explain this, we split the answer.

1. Pseudo-species: A correlation function is a second order function, which means that only two signals are correlated. Therefore, - as we have more than two species - the signal must be “split” (given the selected filters) into two components. Hence, the name of pseudo-species, because at least one pseudo-species is no longer a representation of the signal of a single species. The selection is arbitrary and to a large extent not important, as long as the fluorescence properties of the pseudo-species represent a linear combination of the experimental fluorescence histograms (Eq. S26). Because there is not a unique solution, one could optimize the correlation function to excerpt maximum contrast in the species cross-correlation. In our case, we have a mix of C1 and C2 corresponding to the “major” populations (pseudo-species) and a “minor” (C3) population. Thus, we decided to use these as pseudo-species.

2. Relaxation times: As mentioned on the previous point, the relaxation times correspond to the eigenvalues of the transition rate matrix in a kinetic network. That translated in a correlation function as a summation of exponentials (Eq. S28) with characteristic times as the relaxation times. Thus, a simple minimization algorithm would be useful to determine the temporal response.

3. Amplitudes: The summation of exponentials (Eq. S28) contain an amplitude and a characteristic time. The amplitudes are a function of the reaction rate constants, the brightness of the species or pseudo-species (namely the number of photons), and the FRET-efficiency. Thus, when considering pseudo-species, the amplitudes loose pure meaning as they become a mixture of other parameters that are not possible to disentangle. Fortunately, we have access to the relevant information through other means; namely TCSPC.

In short, when dealing with pseudo-species, the characteristic times are properly distinguished, but the amplitudes are a non-linear combination of many different parameters.

For more details, we have two manuscripts detailing the fundamentals and applications of fFCS and they are referenced.

(31) Felekyan, S., Kalinin, S., Sanabria, H., Valeri, A. & Seidel, C. A. M. Filtered FCS: species auto- and cross-correlation functions highlight binding and dynamics in biomolecules. *ChemPhysChem* **13**, 1036-1053 (2012).

(42) Felekyan, S., Sanabria, H., Kalinin, S., Kühnemuth, R. & Seidel, C. A. M. Analyzing Förster resonance energy transfer with fluctuation algorithms. *Methods Enzymol.* **519**, 39-85 (2013).

3. Review of NCOMMS-18-17509eTCSPC: When the instrument response is subtracted, there are basically no features in the curve. Although the residuals look perfect, I would expect that a multitude of solutions with three relaxation times will fit the data as well. Is it possible to give confidence intervals for the fit parameters?

We agree with the reviewer, therefore, we reported the confidence intervals to our fit parameters; although we bulk them with an error propagation rule in a single uncertainty parameter. For this revision, we split the uncertainty on two tables. One contains the confidence interval (2σ) on the fitting, and the other one contains the uncertainty given κ^2 distribution (Tables S2D-G). We apologize if our notation was not clear enough on the initial submission.

4. eTCSPC data: If I understand this correctly, there is no example where a single relaxation time is sufficient to describe the data, which worries me. If T4L is trapped in a single (fixed) state, there should be only a single relaxation time (distance)?

To explain this further we split into different topics.

1) Time resolved fluorescence of Donor Only variants

It is not uncommon that conjugated dyes show multi-exponential behavior. For example, the donor-only labeled samples already shows a two-exponential decay. Each correspond to different energy levels of the donor.

2) Time resolved fluorescence of Donor in presence of Acceptor

When the donor is the presence of acceptor, the donor lifetime is quenched via FRET. Thus, as the reviewer pointed out, in the ideal case where the donor and acceptor dyes are "fixed" in space and the characteristic energy transfer is determined by the rate of energy transfer -following Förster theory (k_{FRET}), the number of fluorescence lifetimes on the donor-acceptor variant will be the number of donor lifetimes.

However, if in solution, a molecule samples multiple FRET states, the model function should incorporate the fluorescence decay of the donor (i.e. bi-exponential behavior) and consider that a distribution of FRET rates ($p(k_{\text{FRET}})$) acts on each donor lifetime. Therefore, to incorporate this

description, we must use the reference sample (the Donor Only) as reference and extract the proper $p(k_{\text{FRET}})$.

3) Locking configuration and time resolved analysis

According to the TCSPC analysis, the DA-variant, when “locked” into a particular enzyme state, as evidenced by HPLC, shows a distribution of FRET states. This means, that the enzyme fluctuates between different configurations with different relaxation times. Thus, we conclude that the enzyme in solution is not “frozen/fixed” in a particular configuration, even though enzymatically corresponds to the selected state. Biomolecules are intrinsically dynamic due to Brownian motion; thus, the expectation that once a molecule reaches a particular enzymatic state will be frozen in a single configuration is unrealistic when measuring in solution.

In addition, how are the three conformations assigned to the three decay times to start with? Is this assignment left completely open to start the global analysis?

Generally speaking, the implemented algorithm relies on the addition of FRET-states or k_{FRET} 's. Each k_{FRET} corresponds to a different conformation, because FRET quenching of the donor is distance dependent. Thus, we do a systematic approach of adding states. We considered 1, 2, 3, and 4. Then we compare the figure of merit (χ_r^2) for significant improvement between models. When there is no longer statistical justification of addition of parameters, we stop. In our case, we stopped with three.

In addition, we should consider whether the species fraction of each FRET-state is adjusted individually for each variant or globally by a shared species fraction across the full set of variants. We rely on the global determination of species fraction in order to assign FRET-states to a structural conformation. Otherwise, there is no unique way to assign distances to conformational states. Nonetheless, this occurred AFTER we obtained the fit results of the different fit models (Table S5). Therefore, the assignment is left open at the beginning of the minimization and it is given solely by the amplitudes during the optimization algorithm.

Finally, the authors derive a free energy landscape for the three states. As far as I understand, this analysis completely relies on the assumption of a three-state system with a linear succession of the states C3, C2 and C1.

Yes, as reviewer states, our model relies on the linear succession of states. And thanks to the reviewers, this section might be clearer after the revision.

For the details on enzyme function even a certain model (eMMm) is assumed and all results concerning catalysis are only valid if this model is correct. These are strong assumptions and many other (especially larger) models would also fit the data well. Therefore, this has to be clarified and emphasized throughout the text. In particular:

We modified the wording stating clearly our assumptions. For example, after Equation 1 in page 24. We added.

“This model considers a sequential succession of steps to go from a free enzyme to a product bound state.”

In other instances (Page 25), we emphasized our assumptions.

5. Page 25: I cannot see how to observe a certain sequence of states from the presented data. This is something that has been put into the model. In my opinion, here you only get as a result what you already have put in!

On page 25, we mostly describe Fig 7 by explaining the sequence of events along the eMM model. The grayed boxes on 7A represent the major populations as derived by our TCSPC analysis. The major states lie along the diagonal, but because no discontinuous transitions are allowed between reaction states, one moves down the reaction state on the same configurational state. This is better represented in 7B, where the eMM model is connected to the matrix representation.

As a clarification, this sequence is not something we introduced in our eMM model. We treated each sample at different reaction states as independent samples, sharing the same limiting states in order to look at changes in the population of states. Hence, the relative amplitudes show the closing path along the enzymatic reaction. Finally, the eMM model (Fig 7B) is a result of this matrix (Fig 7A), given 3 conformational states (C1, C2, C3), and 3 reaction state (E, ES, EP).

6. Page 28: "a detailed enzymatic reaction scheme is developed": No - this is assumed.

Based on the previous point, we introduced our assumptions but we also mention that the eMM model (Fig. 7B) is the result of the changes in amplitudes along the configuration and enzymatic reaction steps. We hope that by clarifying page 25, we were able to emphasize what is assumed and what is results. Nonetheless, we changed the statement in page 28

"A detailed enzymatic reaction scheme is developed to relate the fluorescence states to the three reaction states and provide a meaningful description of the data."

to:

"Given successive closing of three conformational states (C1, C2, C3), and 3 reaction states (E, ES, EP), we considered a eMM reaction scheme (Fig. 7B) to provide a meaningful description of the data."

7. Kinetic network: I do not understand the argument of "microscopic reversibility of transitions" to discriminate the full cycle. A cycle can also be reversible.

Indeed, as the reviewer states, a cycle can also satisfy microscopic reversibility as long as there are reversible reactions; thus, shunning flux in a particular kinetic network. We don't use the argument of microscopic reversibility (as those words are not found in the manuscript) of transitions, nonetheless, we use the argument of sequential closing of the enzyme (Supplementary Note 3 page 36). A simple analogy is a closing door. The door can go from open to close only passing through states where the door is open at lesser degrees. In order for the door to open again it goes through the intermediate states, as long as there is a wall that limits a different path. The same can be true in the closing of an enzyme. The direct opening from a close state would require that the enzyme unfolds or goes through another set of intermediate states (or through a wall in the door opening analogy) that are not observed. We revised the wording in SI to avoid this confusion.

8. I do not understand the argument for the transition C1-C3 being forbidden (Supplement 2.6.2., now Supplementary Note 3), maybe this can be clarified, because it is a very important point in the argument for the chosen model. In particular, as fFCS shows at least two transition times faster than 10ms (this could also be three) and why is C3t static?

The argument of considering an unlikely direct transition (C1-C3) is given on the previous point. As previously pointed out, we emphasized this assumption.

Regarding the definition of what defines a state "static" (e.g. C3t), we expanded the Supplemental Note 3.2:

“C3t is static because it is accumulated in MFD histograms over the observation time (~ms), and the population of this state lies over the static FRET line. We note that C3t does not increase over the period of the experiment. Hence, this state must be in slow equilibrium (>10 ms) with the rest of the network as reflected by fFCS.

Minor comments:

1. Page 8: How does MFD compare to NMR relaxation dispersion measurements - I cannot find this information in the Supplementary Information as indicated.

The reference to SI 1.1-1.3 is not to relate the NMR dispersion, but rather to provide details on the FRET lines. We moved the reference to Si as follows:

“In MFD-histograms, FRET-lines (Supplementary Information 1.1-1.3) serve as a unique guide to visualize conformational dynamics by peak shifts and splitting like in NMR relaxation dispersion measurements”.

2. Page 25: The authors use the Arrhenius equation with a pre-factor of 10e-03 ms. I have not seen that before, usually 10e-05 ms or less are used (e.g. Joe Howard: "Mechanics of Motor Proteins and the Cytoskeleton"). In addition, I suggest to use the unit sec or Hz.

According to what is stated by Joe Howard “the Arrhenius theory provides no information about the frequency factor, A”. Two models exist: the Eyring and the Kramer’s rate theory. In our case, the pre-factor (A) in Arrhenius equation is chosen as to provide activation energies that are reasonable with thermally driven processes; hence, selected accordingly. Until we find a benchmark system to relate the kinetics in the absence of an external force with a well-defined reaction coordinate, this approach should be taken as a semi quantitative description to compare the energy barriers between states.

The use of ms is consistent with the notation of the reported rates of T4L found in published manuscripts reporting rate constants in ms^{-1} . As the reviewer might be more familiar with Hz or s, other people are more familiar with the use of ms^{-1} . In either case, we will always fail to satisfy everyone. I hope the reviewer understands.

3. Page 26: I do not understand the word "processivity" in the context of this enzymes. Please explain.

T4L is an enzyme whose main purpose is to catalyze the peptidoglycan in order to disrupt the bacterial cell wall. Thus, in the absence of a driving force to free the enzyme, T4L might get stalled due to the excess of substrate. Processivity in this case, indicates the directionality of the reaction that goes from a state where there is an excess of Substrate and absence of Product to a state where there is excess of Product and absence of Substrate.

To avoid confusion, we changed the word processivity to **directionality**.

4. Fig. S2B: I cannot see a “more pronounced tilt towards the C3”. What I see is a broadening of the peak, which also could be caused by a tilt towards C2.

As future reference, we use the word broadening as changes in width over a single parameter of one dimension, we use tilting where we are discussing two dimensions or simultaneous changes in two parameters. A tilt implies an angle, thus referring to two dimensions. Broadening over F_D/F_A or broadening over the average fluorescence lifetime would correspond to a different behavior.

To avoid confusion, we rewrote it like:

“more pronounced broadening along the FRET-line in direction to C3”

Reviewer 3:

This evaluation refers only to the DEER results of this manuscript

The authors use DEER measurements of one of their many mutants to support the FRET based finding of the existence of three conformations. It is well known, that distance distributions derived using Tikhonov regularization often show some shoulders and minor peaks in the distance distributions and this often depends on the regularization parameter chosen for the analysis. Therefore, the assignment to different conformations of the protein is highly questionable (can be also originate from the label) unless some additional systematic experiments are carried out to substantiate such assignment. The authors should carry out validations available in the DEERanalysis software to highlight uncertainties, these are usually pretty good in revealing what features in the distribution can be trusted. These, together with the primary DEER data should be added to the SI.

We thank the reviewer for critically reviewing the EPR results. We agree with the reviewer that proper error analysis must be taken. **Unfortunately, the primary data for the pH 7.0 (typo on the text referring to it as pH 7.5) was not available. These experiments were collected on 2011 (8 years ago). Due to changes in personnel, hard-disks, and naming files, we were able to locate only the data corresponding to the pH 3.0 and the T26E conditions (Figures at the end), but we couldn't locate the data at pH 7.0. For this reason, we no longer report our data on the pH 7.0.**

From the error analysis of the identified files, it is possible to see that the uncertainties along the interspin distance interval are minimal. A spurious but significant error was identified beyond 8 nm (See figure below). Nonetheless, this regime was not considered in the first submission. Now, the PELDOR/DEER spectra are shown in Figure 6, Figure S5D, and shown in the figure at the end of the response, with the corresponding fit.

I scanned the literature and there are many papers showing DEER measurements on a variety of double mutants of T4L using different types of nitroxide spin labels and even a few Gd(III) based spin labels. All these usually show one peak in the distance distribution. Particularly narrow distance distributions are obtained with double arm labels (both nitroxide and Gd(III)). In the case of the nitroxide the comparison with the usual R1 label demonstrate well the effect of the label in the broadening. Although, the labeling sites are not exactly the same as in this paper they are not too far away and I would expect to see the different conformations there but they do not even show two resolved conformations. This should be discussed.

Islam S.M. *et al* (J. Phys Chem B, 117(17), 4740, 2013) carried out 37 different DEER experiments on T4L. The authors show that some interspin distributions are somewhat heterogenous, some show broader distributions, others show satellite peaks reminiscent of our data, but the majority as the reviewer noticed are “narrow” distributions. Also, Stein R.A. *et al* (Meth. in Enzym. 563:531, 2015) uses T4L for introduction of the DEER analysis. However, in those studies the sites 44/150 were not used. Also, we would like to note that the experimental conditions were slightly different. Due to the complexities on retrieving the original dataset, we no longer present the interspin distance distribution of the pH 7.0 data. Thus, we focus our response to the data that is currently

presented on the manuscript. In that respect, we added the proper citation and reference to these published data set.

Having said that, the important message of the EPR experiments is that they support the view of the existence of a more compact state at shorter interspin that increases in population at pH 3.0 (See Figure below) analogous to what was observed in FRET experiments.

I would interpret the DEER data more cautiously as follows: the pH=7.5 has a broad distance distribution which, in addition to the spin label induced broadening (it can actually be estimated from the crystal structures using MMM) reflects the flexibility of the protein and represents a family of conformations. The linked construct restricts the dynamics of the protein leading to a narrower distance distribution. The change of pH leads to a similarly broad distribution as the 7.5 pH sample, with but with a shorter mean distance, indicating that different conformations are present. I am really not convinced that these data show that three well distinct conformations are present in the pH=7.5.

We agree with reviewer's point of view. EPR cannot resolve all distinct states. Because we no longer present the pH 7.0 data, we remove all possible interpretation of that data.

Minor comments

Better use the name MTSSL and not HO-225 for the nitroxide agent used to label the cysteine.

Changes made accordingly.

Why is the distance distribution cut a little below 3 nm, cutting the dashed distance distribution? Please show the distance distribution starting at 1.5 nm.

Distance distributions were computed from 2 nm (Figure 6, Figure S5D, and Figure below).

Distance Distribution T4L 44R1/150R1 pH 3.0

Distance Distribution T4L 44R1/150R1 T26E Adduct

Figure. Error analysis of DEER data for the pH 3.0 and the T26E adduct.

REVIEWERS' COMMENTS:

Reviewer #2 (Remarks to the Author):

All my points have been addressed in the point by point response. In particular the authors clarified several parts of their analysis procedure and their assumptions.

Unfortunately, not all explanations found their way into the manuscript, maybe some more can be added in the final version, e.g.:

(1) In Fig. 1 of the response helpful dashed lines are included, which did not make it into the manuscript.

(2) The details on the pseudo-species (around equation S26) are still not fully clear to me. I also cannot find the important information that the selection was arbitrary.

(3) The explanation of why the transition C1-C3 is not considered (or unlikely) could be strengthened in the main text. This is a crucial point, as the efficiency vs lifetime plots show lines directly connecting C1 and C3 (with some data falling on these lines). In addition, the correlation between the conformational states and the reaction states (Fig. 7A) is not very strong, therefore C2 could be "on pathway" or "off pathway" (as C1 and C3 are also significantly populated in the ES state).

Altogether I am convinced that the findings are novel and should be of interest to others in the community and have a large potential to influence thinking in the field. Therefore, I support publication of the revised manuscript in Nature Communications.

REVIEWERS' COMMENTS:

Reviewer #1 (Remarks to the Author):

No comments

Reviewer #2 (Remarks to the Author):

We appreciate the reviewer for going over the proposed changes and the favorable response. Based on these remarks we made changes as suggested by the reviewer.

All my points have been addressed in the point by point response. In particular the authors clarified several parts of their analysis procedure and their assumptions.

Unfortunately, not all explanations found their way into the manuscript, maybe some more can be added in the final version, e.g.:

(1) In Fig. 1 of the response helpful dashed lines are included, which did not make it into the manuscript.

We replaced figure 1 on manuscript to include the dashed lines.

(2) The details on the pseudo-species (around equation S26) are still not fully clear to me. I also cannot find the important information that the selection was arbitrary.

We added the word "arbitrary" in page 23 of main text (Highlighted in cyan). Moreover, in page 49 of the supporting information we added the following sentences for clarification:

"Here, $p_j^{(i)}$ corresponds to the time resolved fluorescence decay of each selected pseudo-species i (Supplementary Figure 9a,c). Using the definition of the fluorescence decays in Equation 26 it is then possible to find a weight value $w^{(i)}$ to satisfy the experimental observable fluorescence decay characteristic of the mixture H_j , and the corresponding filters per species $f_j^{(i)}$. Examples are plotted in Supplemental Figure 9B and 9D".

(3) The explanation of why the transition C1-C3 is not considered (or unlikely) could be strengthened in the main text. This is a crucial point, as the efficiency vs lifetime plots show lines directly connecting C1 and C3 (with some data falling on these lines). In addition, the correlation between the conformational states and the reaction states (Fig. 7A) is not very strong, therefore C2 could be "on pathway" or "off pathway" (as C1 and C3 are also significantly populated in the ES state).

Although some bursts fall in the connection between C_1 and C_3 , in order to satisfy population fractions in equilibrium an effective transition from C_1 to C_3 should be equal to C_1 - C_2 - C_3 . Thus, bursts will be split and that is not observed. Moreover, the Monte Carlo simulations, considering the linear connectivity, would not have agreed with the experimental observations. Thus, as

suggested by the reviewer, we included the next sentences into the manuscript to strengthen why the C_1 - C_3 transition is unlikely.

Page 9 “while the cycle scheme $C_1 \rightleftharpoons C_2 \rightleftharpoons C_3 \rightleftharpoons C_1$ is unlikely due to the lack of bursts across all FRET variants that connect $C_3 \rightleftharpoons C_1$ with an effective slower rate to satisfy equilibrium. These bursts would follow the dynamic FRET-lines as guides between states in the MFD-diagram (Figure 3a and Supplementary Figure 2).”

Supplementary Note 3

Page 39 “due to the sequential closing of the enzyme from the most open to the most closed state, due to the unseen number of slow transitions along the C_3 to C_1 , and physical restrains observed in the structural models (PDB ID 172L and 148L)”

Moreover, we added the following sentence to discuss about the difference of the “on pathway” to that of the “off pathway”

Page 19 “which is consistent with our Monte Carlo simulations and incompatible with the unlikely off pathway from C_1 to C_2 through C_3 due steric hindrances and by the fast hinge bending motion ($4 \mu\text{s}$) expected from structural models (PDB ID 172L and 148L).”

Altogether I am convinced that the findings are novel and should be of interest to others in the community and have a large potential to influence thinking in the field. Therefore, I support publication of the revised manuscript in Nature Communications.

Last, we added a recent citation regarding the stability of the helix c of T4L. Reference:
52. Xue, M. et al. How internal cavities destabilize a protein. Proc Natl Acad Sci U S A 116, 21031-21036 (2019).